EMBO
Molecular Medicine

# Systematic pharmacological screens uncover novel pathways involved in cerebral cavernous malformations

Cécile Otten[1], Jessica Knox[2,3,4,†], Gwénola Boulday[5,†], Mathias Eymery[6,7,8], Marta Haniszewski[2,9], Martin Neuenschwander[10], Silke Radetzki[10], Ingo Vogt[11,12], Kristina Hähn[1], Coralie De Luca[5], Cécile Cardoso[5], Sabri Hamad[11,12], Carla Igual Gil[1], Peter Roy[2,3,4], Corinne Albiges-Rizo[6,7,8], Eva Faurobert[6,7,8], Jens P von Kries[10], Mónica Campillos[11,12], Elisabeth Tournier-Lasserve[5,13], W Brent Derry[2,9] & Salim Abdelilah-Seyfried[1,14,*] (iD)

## Abstract

Cerebral cavernous malformations (CCMs) are vascular lesions in the central nervous system causing strokes and seizures which currently can only be treated through neurosurgery. The disease arises through changes in the regulatory networks of endothelial cells that must be comprehensively understood to develop alternative, non-invasive pharmacological therapies. Here, we present the results of several unbiased small-molecule suppression screens in which we applied a total of 5,268 unique substances to CCM mutant worm, zebrafish, mouse, or human endothelial cells. We used a systems biology-based target prediction tool to integrate the results with the whole-transcriptome profile of zebrafish CCM2 mutants, revealing signaling pathways relevant to the disease and potential targets for small-molecule-based therapies. We found indirubin-3-monoxime to alleviate the lesion burden in murine preclinical models of CCM2 and CCM3 and suppress the loss-of-CCM phenotypes in human endothelial cells. Our multi-organism-based approach reveals new components of the CCM regulatory network and foreshadows novel small-molecule-based therapeutic applications for suppressing this devastating disease in patients.

Keywords angiogenesis; CCM; ERK5; indirubin-3-monoxime; KLF2

Subject Categories Cardiovascular System; Pharmacology & Drug Discovery; Vascular Biology & Angiogenesis

## Introduction

Cerebral cavernous malformations are characterized by the presence of vascular lesions that are most prevalent in the brain and can occur sporadically or through a familial condition from mutations in CCM1/KRIT1, CCM2/Malcavernin, or CCM3/PDCD10 [reviewed in (Chan et al, 2010; Riant et al, 2010; Fischer et al, 2013)]. The loss of CCM3 has particularly been associated with an early onset and severe progression of the pathology (Riant et al, 2013; Shenkar et al, 2015). Surgical resection is currently the only curative option for CCM patients, but when lesions are seated deeply in regions of the brain not accessible to neurosurgery, the condition may cause severe morbidity or even lethality due to recurrent cerebral hemorrhages. Hence, pharmacological interventions are desperately needed to (i) prevent the growth and bleeding of existing lesions and (ii) to suppress the formation of new ones.

---

1   Institute of Biochemistry and Biology, Potsdam University, Potsdam, Germany
2   Department of Molecular Genetics, University of Toronto, Toronto, ON, Canada
3   The Donnelly Centre for Cellular and Biomolecular Research, University of Toronto, Toronto, ON, Canada
4   Department of Pharmacology and Toxicology, University of Toronto, Toronto, ON, Canada
5   INSERM UMR-1161, Génétique et physiopathologie des maladies cérébro-vasculaires, Université Paris Diderot, Paris, France
6   INSERM U1209, Grenoble, France
7   Institute for Advanced Biosciences, Université Grenoble Alpes, Grenoble, France
8   CNRS UMR 5309, Grenoble, France
9   Developmental and Cell Biology Program, The Hospital for Sick Children, Toronto, ON, Canada
10  Leibniz-Forschungsinstitut für Molekulare Pharmakologie, Berlin, Germany
11  German Center for Diabetes Research, Neuherberg, Germany
12  Institute of Bioinformatics and Systems Biology, Helmholtz Zentrum München, Neuherberg, Germany
13  AP-HP, Groupe hospitalier Saint-Louis, Lariboisière, Fernand-Widal, Service de génétique moléculaire neuro-vasculaire, Paris, France
14  Institute of Molecular Biology, Hannover Medical School, Hannover, Germany
    *Corresponding author. Tel: +49 3319775540; E-mail: salim.seyfried@uni-potsdam.de
    †These authors contributed equally to this work

Loss of CCM proteins in *Caenorhabditis elegans*, zebrafish, and mouse leads to penetrant defects in vascular structures and conserved signaling pathways. Importantly, these model organisms offer many advantages for suppressing these phenotypes using small molecules or genetic methods. In the nematode *C. elegans*, complete loss of the *CCM1* homologous gene *kri-1* causes resistance to apoptosis (Ito *et al*, 2010), whereas loss of *ccm-3* causes defects in biological tube development (Lant *et al*, 2015; Pal *et al*, 2017). Similarly, loss of any of the Ccm proteins in zebrafish and mouse causes cardiovascular malformations that result in cardiac defects, including abnormal cardiac chamber ballooning, a failure of endocardial cushions to form at the atrioventricular canal, and defects in blood vessel formation (Mably *et al*, 2003, 2006; Hogan *et al*, 2008; Boulday *et al*, 2009; Kleaveland *et al*, 2009; Zheng *et al*, 2010; Yoruk *et al*, 2012; Renz *et al*, 2015; Zhou *et al*, 2015). In mice, the endothelial-specific deletion of *Ccm1-3* at postnatal day 1 leads to lesions in the CNS and retinal vasculature which resemble CCM lesions in patients (Boulday *et al*, 2011).

Several molecular pathways have been implicated mechanistically in the pathological changes that occur within endothelial cells upon the loss of CCM proteins. These include signaling via transcription factors KLF2/4 (Maddaluno *et al*, 2013; Renz *et al*, 2015; Zhou *et al*, 2015, 2016), the innate immunity TLR4 receptor (Tang *et al*, 2017), MAPK (Uhlik *et al*, 2003; Fisher *et al*, 2015; Zhou *et al*, 2015, 2016; Cuttano *et al*, 2016), β1-integrin (Brütsch *et al*, 2010; Faurobert *et al*, 2013; Renz *et al*, 2015), angiogenesis and/or Notch (Boulday *et al*, 2009, 2011; Brütsch *et al*, 2010; Wüstehube *et al*, 2010; Zhu *et al*, 2010; You *et al*, 2013, 2017; Renz *et al*, 2015; Schulz *et al*, 2015; Lopez-Ramirez *et al*, 2017), Rho/ROCK (Glading *et al*, 2007; Whitehead *et al*, 2009; Borikova *et al*, 2010; Stockton *et al*, 2010; Richardson *et al*, 2013), and BMP/TGFβ/endoMT or Wnt (Maddaluno *et al*, 2013; Bravi *et al*, 2015, 2016). In addition, the pathology may be accompanied by increases in the production of ECM-degrading metalloproteases (Zhou *et al*, 2015), the secretion of angiopoietin-2 (Jenny Zhou *et al*, 2016), and oxidative stress (reviewed in Retta & Glading, 2016), or defective autophagy (Marchi *et al*, 2015) and apoptosis (Ito *et al*, 2010). The identification of these misregulated signaling pathways has suggested potential routes for pharmacological interventions, and molecules that modulate these pathways have been tested in preclinical studies of murine endothelial-specific inducible CCM models and even in clinical studies. These studies investigated the potential of the Rho/Rock signaling inhibitors fasudil or simvastatin (Zhou *et al*, 2015; Shenkar *et al*, 2017), the blood pressure lowering drug propranolol (Reinhard *et al*, 2016), the Wnt signaling inhibitor sulindac sulfone (Bravi *et al*, 2015), the TGFβ and pSMAD inhibitors LY-364947 and SB-431542 (Maddaluno *et al*, 2013), the TLR4 antagonist TAK-242 (resatorvid; Tang *et al*, 2017), or treatment with antibiotics (Tang *et al*, 2017). However, since many of these candidate drugs may cause severe side effects in patients and because a comprehensive overview of CCM-relevant molecular pathways is still lacking, the scientific community has recognized the need of performing more systematic small-molecule screens. The fastest route to clinical trials for drugs in the treatment of CCMs would be repurposed substances already on the market. Recently, two screens based on Food and Drug Administration (FDA)-approved small molecules have been performed: the first, carried out on human CCM2-deficient

endothelial cells, led to the identification of several molecules including tempol, a free radical scavenger, and vitamin D, neither of which had previously been implicated in the treatment of this disease (Gibson *et al*, 2015). Another screen assayed CCM3-deficient mouse primary astrocytes and *Drosophila* glial cells for the suppression of overproliferation. This led to the identification of compounds affecting the mevalonate pathway (Nishimura *et al*, 2017). While these studies have provided important first insights into potential therapeutic approaches, an unbiased, integrative, and multi-organismic screen has not previously been attempted. Performing such compound screens in the context of the complex multi-tissue comprising CCM-deficient organisms may provide comprehensive insights into conserved druggable pathways. Vertebrate models with a cardiovascular system such as zebrafish offer additional advantages that may not be available by screening cultured cells or invertebrates. Hence, combined screens using multiple systems are more likely to provide a comprehensive list of CCM-relevant compounds.

Here, we present the results of a repurposed drug screen that assayed the efficacy of suppressing cardiovascular defects in zebrafish *ccm2$^{m201}$* mutants or synthetic lethality in *kri-1*; *ccm-3* double mutants in *C. elegans*. Our study combines system biological analyses integrating *ccm2$^{m201}$* mutant transcriptional data with molecular pathways that have been modulated using small-molecule compounds. These analyses pinpoint particular disease signatures as critical hubs that could be targeted by therapies. In addition to many previously identified compounds, our unbiased screen provides a range of new candidates that affect angiogenesis, vitamin D and retinoic acid signaling, blood pressure, ion channels, neurotransmitters, the oxidative stress/redox system, inflammation, and the innate immune system. These findings provide an unbiased framework for therapeutic approaches to tackle this debilitating disease. The relevance of this unbiased screen for CCM therapeutics is well illustrated by the identification of indirubin-3-monoxime as a compound showing a rescue in human endothelial cells and a strong preventive effect in CCM mouse models.

## Results

### Repurposed drug screens identify compounds that suppress CCM mutant phenotypes in zebrafish and *C. elegans*

Most screens in the past have been primarily based on simplified *in vitro* models that had only a limited ability to recreate the complexity of the cardiovascular system or of the complex whole organismal interactions that may be affected in the CCM pathology (Gibson *et al*, 2015; Nishimura *et al*, 2017). To identify compounds for a pharmacological suppression of CCM phenotypes, we employed diverse assays on multiple organisms that can help to discriminate distinct effects on the cardiovascular system or cell biology upon loss of CCM proteins (Fig 1A). These assays included screening small compound libraries at concentrations of 10 μM for 24 h in zebrafish *ccm2$^{m201}$* mutant embryos carrying the endothelial-specific reporter transgene Tg (*kdrl: GFP*)$^{s843}$ and probing for the suppression of the ballooning heart phenotype at 48 h postfertilization (hpf) (Mably *et al*, 2006; Materials and Methods). In parallel, we took advantage of the

                                    

**Figure 1.   Small-molecule drug screens identify compounds relevant for CCM.**

A     Overview of the four different screening assays used in this study. Zebrafish embryos and *C. elegans* are screened in 24-well and 96-well plates, respectively. The most promising active compounds are retested in *shCCM2* HUVECs. One compound is tested for suppression of vascular lesion formation in the cerebellum of *iCCM2* and *iCCM3* mouse models.

B     Overlap of rescue compounds screened in the different assays.

C–E   Examples of rescue of cardiovascular defects of the zebrafish $ccm2^{m201}$ mutant. Inverted images of confocal z-scan projections of the 46 hpf head region and heart (endocardium) of wild-type (WT) and $ccm2^{m201}$ mutant zebrafish embryos carrying the endothelial $Tg(kdrl:GFP)^{s843}$ reporter transgene. Embryos are untreated (C) or treated between 17 and 48 hpf with 10 µM of the Lck inhibitor C8863 (D) or with 10 µM of the ERK5 inhibitor XMD8-92 (E). Both compounds resulted in a reduction in heart size and narrowing of the heart tube at the atrioventricular canal (arrowheads). Scale bar is 100 µm.

synthetic lethality caused by the co-ablation of *kri-1* (*CCM1*) and *ccm-3* in *C. elegans* (Lant *et al*, 2015; Materials and Methods) and screened for a restoration of viability, which led to insights into cellular processes affected by the loss of CCM proteins that are conserved between the species.

We screened a total of 1,600 unique compounds in zebrafish (LOPAC/Selleck libraries), 8.4% of which (134/1,600) alleviated the $ccm2^{m201}$ mutant heart phenotype (Fig 1B–E; Dataset EV1). Concurrently, we screened 4,748 unique compounds [LOPAC/Selleck, Spectrum, and GlaxoSmithKline protein kinase inhibitors (GSK-PKIs)] in *C. elegans kri-1*(*ok1251*) mutants fed *ccm-3* RNAi, 7.4% of which (350/4748) rescued the synthetic lethal phenotype (Figs 1B and EV1; Dataset EV1). The two screens identified six compounds that had already been implicated in alleviating CCM loss-of-function phenotypes in other models: sulindac sulfone (Bravi *et al*, 2015), XMD8-92 (Cuttano *et al*, 2016), cholecalciferol (Gibson *et al*, 2015), propranolol hydrochloride (Moschovi *et al*, 2010), simvastatin

(Whitehead *et al*, 2009), and sorafenib tosylate (Wüstehube *et al*, 2010; Table EV1).

Among the 1,080 compounds that were screened in both zebrafish and *C. elegans*, 32 suppressed CCM phenotypes in both models (Fig 1B; Datasets EV1 and EV2). This overlap is highly significant (*P*-value = 0) when compared to a random scenario. Finally, we used human umbilical cord venous endothelial cells (HUVECs) treated with *CCM2* shRNA to screen 31 compounds that showed rescue both in zebrafish and *C. elegans* and another 131 compounds that had been particularly effective in one animal model or the other. Of these 162 compounds, 26 rescued at least some features of the *CCM2* phenotype, which is characterized by an increase in the formation of stress fibers, reduced levels of cortical ACTIN, and cell shapes that are more elongated than control shRNA-treated HUVECs (Faurobert *et al*, 2013; Materials and Methods; Fig 1B; Datasets EV1 and EV2). For example, indirubin-3-monoxime (IR3mo), which was identified in the zebrafish screen, also gave a rescue in the HUVECs

screen (Datasets EV1 and EV2). Five compounds showed some degree of rescue in all three CCM models (Dataset EV2): the FLT3 angiogenesis inhibitor ENMD-2076, the PKC/phospholipase A2/D inhibitor DL-erythro-dihydrosphingosine, the PI3K/Akt/mTor pathway inhibitor ridaforolimus, the muscarinic acetylcholine receptor antagonist DL-homatropine hydrobromide, and 13-cis-retinoic acid, which has anti-inflammatory and anti-tumorigenic effects. Strikingly, most of the molecular pathways targeted by these small molecules had not previously been implicated in CCM.

**Classification of compound activities**

A functional annotation analysis based on the 18% (24/134) of the zebrafish-active compounds and 20% (70/350) of the *C. elegans*-active compounds with at least one Medical Subject Headings (MeSH) term assignment revealed that they have a wide range of therapeutic uses (Fig EV2), physiological effects (Fig EV3A), and affect distinct molecular mechanisms (Fig EV3B). The comparison of 102 different therapeutic, physiological, or molecular terms according to which the compounds active in zebrafish and *C. elegans* were classified revealed an enrichment of anti-inflammatory, anti-hypertensive, neurotransmission modulatory, anti-oxidative, or anti-neoplastic functions. Importantly, we found a number of examples where different compounds with a shared pharmacological functional annotation alleviated CCM phenotypes only in either one or the other animal model. These included vasodilatory agents, for which felodipine (F 9677) affected only zebrafish and carvedilol (C 3993) affected only *C. elegans*, or the sensory system agents for which niflumic acid (N 0630) affected only zebrafish and loxoprofen (L 0664) only *C. elegans* (Table EV2).

**Target protein predictions reveal networks involved in CCM**

Clustering analyses based on MeSH term assignments did not help characterize the potential therapeutic, physiological, or molecular activities of many compounds because their mode of action had not yet been sufficiently defined to be assigned a MeSH term. To predict additional protein targets of active compounds, we applied the DePick computational target deconvolution tool. This draws on an extended version of the human drug target prediction tool HitPick to determine protein targets of small compounds identified in phenotypic screens (Liu *et al*, 2016; Materials and Methods). Based on 1,472 of 1,600 compounds with annotated targets in zebrafish and 4,170 of 4,748 in *C. elegans*, DePick predicted 47 and 134 human proteins as statistically significant targets of the compounds identified in the zebrafish and in the *C. elegans* screens, respectively (Table EV3; Dataset EV3). Several of the targets identified in the *C. elegans* screen had previously been implicated in CCM; these included TLR4 (Tang *et al*, 2017), metalloproteinases (MMP2, MMP7, MMP13, MMP14; Zhou *et al*, 2015), and HMGCR (Nishimura *et al*, 2017), which is a strong validation of the DePick method (Table EV3).

DePick analyses revealed a number of important insights into the regulatory network involved in CCM. First, DePick datasets identified a number of specific processes that were targeted in both zebrafish and *C. elegans* CCM mutants. We carried out comparative Gene Ontology (GO) term analyses for biological processes (GO-BP) based on the zebrafish and *C. elegans* DePick datasets (Ashburner *et al*, 2000; The Gene Ontology Consortium, 2017). We identified 42 significantly targeted GO-BP terms based on the zebrafish DePick dataset; the analysis of the *C. elegans* dataset using these terms revealed that 20 of those were also statistically significant (Dataset EV4). Direct comparison of the zebrafish and *C. elegans* datasets revealed that among the most significantly targeted proteins in both compound screens were nine proteins with a role in vitamin D or retinoic acid signaling (Table EV3; Datasets EV3 and EV4). In addition, GO-BP terms common to the two groups included cell cycle control mechanisms (Dataset EV4). Biological processes exclusively based on the zebrafish DePick dataset included signaling by the PKB, PDGFR, and PI3K pathways (Dataset EV4).

Second, DePick datasets were used to visualize protein interaction networks related to CCM based on the use of "Search Tool for the Retrieval of Interacting Genes/Proteins" (STRING) clustering algorithms for known protein interaction networks (https://string-db.org/). These analyses revealed clusters among the proteins that were present in both the zebrafish and *C. elegans* DePick target datasets including vitamin D signaling proteins (CYP2R1 and CYP24A1) and retinoic acid receptors (RAR and RXR; Figs 2 and 3, and EV4). We also found that some of the most significantly targeted proteins in the zebrafish dataset are clustered in the VEGFR-dependent signaling pathway that controls angiogenesis (KDR, FLT1, and FLT4; Fig 2). Additionally, we identified a protein cluster comprising cytoskeletal and cell cycle-related proteins (tubulins, polo-like kinase PLK1, and cyclin-dependent kinase CDK5), which also was exclusive to zebrafish (Fig 2). In the *C. elegans* DePick targets, we uncovered several protein clusters including one related to lipid metabolism (ALOX5, ALOX12, ALOX15, ALOX15B, and ALOX5AP; Fig EV4).

Third, DePick data helped analyze transcriptional changes related to the loss of CCM proteins. We compared misregulated genes from a previously published transcriptome of zebrafish *ccm2^m201* mutant cardiac tissue (Renz *et al*, 2015) with the zebrafish DePick target dataset (Howe *et al*, 2013) and found a significant misregulation of nine of the 47 genes encoding proteins that were directly orthologous to predicted target proteins. In line with the protein interaction network data, these included the retinoic acid receptor gene *raraa*, the cellular retinoic acid-binding protein genes *crabp1b* and *crabp2b*, and the angiogenesis signaling receptors *flt1*, *kdr*, and *flt4* (Table EV4). To assess whether additional genes related to disease-relevant genetic pathways might be affected, we carried out comparative GO-BP term analyses based on the zebrafish DePick dataset and found that only two GO-BP terms were also significant for the zebrafish *ccm2^m201* mutant cardiac tissue dataset (Ashburner *et al*, 2000; The Gene Ontology Consortium, 2017): In line with evidence from several functional studies, our analyses indicate that VEGF-dependent angiogenesis signaling is relevant to the disease (Wüstehube *et al*, 2010; Zhu *et al*, 2010; You *et al*, 2013; Renz *et al*, 2015; Lopez-Ramirez *et al*, 2017), as well as retinoic acid signaling since several of these genes cluster under the GO-BP term "anatomical structure morphogenesis" (Fig 2; Dataset EV4). Furthermore, we found that the closely related GO-BP terms "cytoskeletal organization" and "regulation of cytoskeletal organization" were relevant for the zebrafish DePick and the zebrafish transcriptome datasets, respectively (Table EV5; Dataset EV4). This finding was in good agreement with the observation that CCM mutant endothelial cells show defective cytoskeletal organization

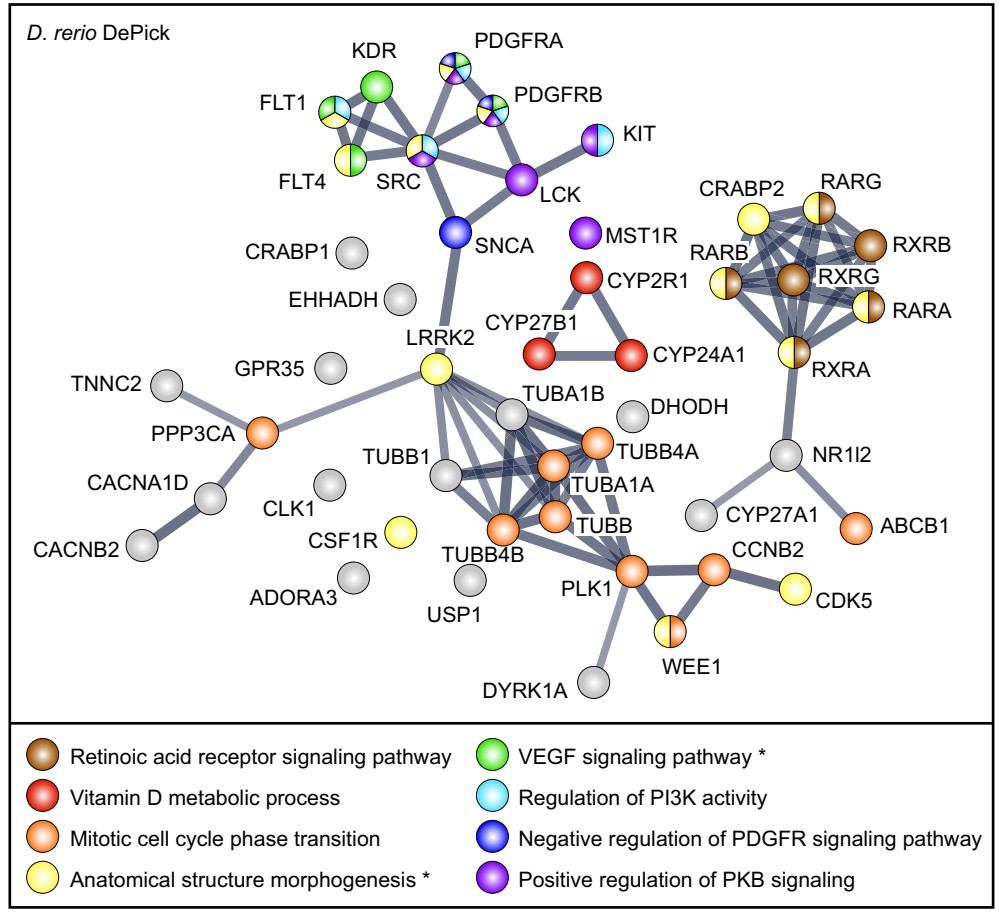

**Figure 2.  STRING network clustering for interactions of zebrafish DePick target proteins.**

Nodes are DePick target proteins, with color coding according to Gene Ontology biological process (GO-BP) terms. GO-BP terms highlighted here are based on DePick target protein predictions of compounds identified in the zebrafish screen. Asterisks indicate GO-BP terms which are also enriched in the zebrafish transcriptome of $ccm2^{m201}$ mutant hearts (Renz et al, 2015). Edges are protein–protein associations. Line thickness represents the strength of data support.

(Glading et al, 2007; Faurobert et al, 2013). In summary, comparative analyses based on DePick target protein predictions have led to the novel discovery of potential drugs that impinge on a set of molecular pathways and biological processes that are affected in both animal models, as well as a few that appear to be species-specific.

**Indirubin-3-monoxime shows rescue activity in different CCM models**

Given the stringent limitation inherent to mouse preclinical trials, we defined a number of criteria as guidelines for compound selection: First, it should be a compound that is widely used in long-term medical treatment plans but exhibits few or no side effects in patients. Second, the compound should interfere with several molecular pathways relevant to CCM. Finally, we ranked the positive effects of a drug in the zebrafish ccm model higher compared to its effects in C. elegans due to the evolutionary conservation of the vertebrate cardiovascular systems. Among the compounds identified in the screen, we selected IR3mo, which suppressed the massive cardiac ballooning phenotype of zebrafish $ccm2^{m201}$ (Fig 4A–D) and

$krit1^{ty219c}$ mutants (Fig EV5A–D), for further tests in preclinical mouse models even though it was not identified in the C. elegans screen.

IR3mo is an FDA-approved drug with low toxicity derived from traditional Chinese medicine that has been widely used to treat leukemia and other chronic diseases (Eisenbrand et al, 2004; Williams et al, 2011). Molecular studies demonstrated that indirubin or its derivate IR3mo inhibits VEGFR2/KDR-dependent angiogenesis (Zhang et al, 2011), among other targets and cyclin-dependent kinases involved in cell cycle regulation (Hoessel et al, 1999). Based on the zebrafish DePick dataset, three of 10 proteins predicted to be targets of IR3mo were statistically significantly targeted in the screen (SRC, VEGFR2/KDR, and MST1R), which suggested that IR3mo may have its effects through its action on several targets (Fig 2; Table EV6). So, even though IR3mo did not rescue the phenotype in C. elegans, we considered it a good candidate for further studies.

We first addressed the issue of whether IR3mo could rescue human endothelial cells and assayed its effects on siCCM1/KRIT1-treated HUVECs, which are characterized by a massive formation of stress fibers and elongated cell shapes (Fig 4E–H). In keeping with a conserved effect in human endothelial cells, IR3mo treatment

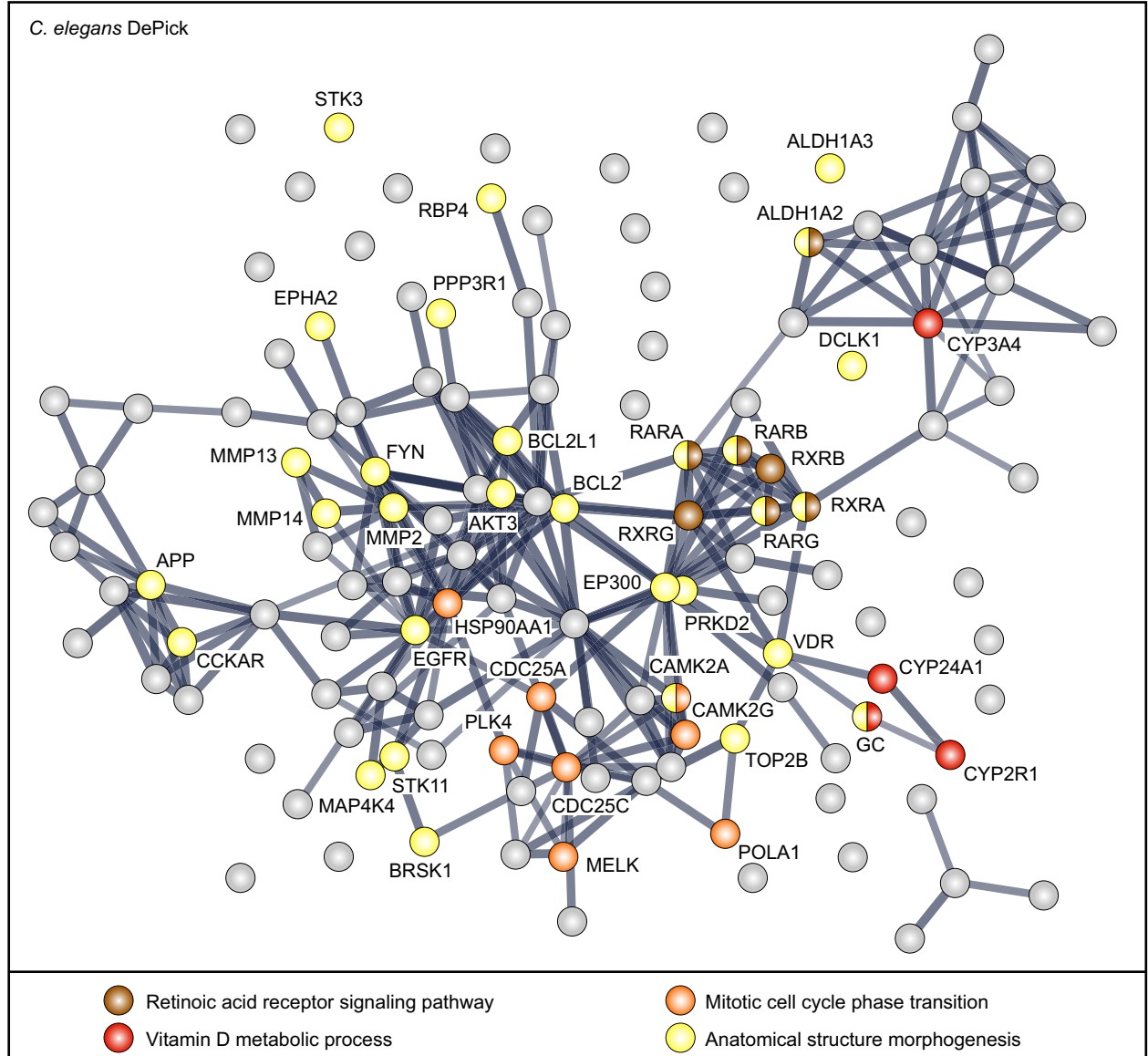

**Figure 3. STRING network clustering for interactions of *C. elegans* DePick target proteins.**

Nodes are DePick target proteins, with color coding according to Gene Ontology biological process (GO-BP) terms. GO-BP terms highlighted here are based on DePick target proteins of compounds identified in the *C. elegans* and zebrafish screens. See Fig EV4 for all DePick target protein names in this figure. Edges are protein–protein associations. Line thickness represents the strength of data support.

efficiently suppressed the cellular tension phenotype, restoring cells to a rounded morphology with ruffled cell membranes (Faurobert *et al*, 2013; Fig 4H). Similarly, *CCM2*- and *CCM3*-silenced HUVECs were rescued by IR3mo treatment (Fig EV5E–J). Because CCM-associated cardiovascular defects in mouse and zebrafish models have been linked to an activation of the MEKK3-MEK5-ERK5 signaling pathway (Zhou *et al*, 2015; Cuttano *et al*, 2016) and elevated levels of *KLF2* (Renz *et al*, 2015; Zhou *et al*, 2015, 2016), we next tested the molecular effects of IR3mo treatment on these phenotypes in different CCM models. Strikingly, IR3mo treatment efficiently abolished the elevated levels of pERK5 in *CCM1/KRIT1*-depleted HUVECs (Fig 4I–K). IR3mo treatment also reduced *KLF2* mRNA to

wild-type levels in *CCM1/KRIT1*-depleted HUVECs, which is consistent with a strong transcriptional activation of *KLF2* by pERK5 (Fig 4L), and also reduced *klf2a* mRNA levels in *ccm2^{m201}* mutant zebrafish embryos (Fig 4M). In summary, these results indicate that IR3mo suppresses the hyper-activation of MAPK signaling and *KLF2* induction upon loss of CCM proteins.

To elucidate the impact of IR3mo in a preclinical murine model mimicking brain CCM lesions of the human brain vasculature, we assayed its efficacy on the lesion burden in inducible endothelial-specific knockout models of *CCM2* and *CCM3* (Boulday *et al*, 2009). The induction of the conditional *CCM2* or *CCM3* knockouts at P1 resulted in a severe form of the disease that is characterized by

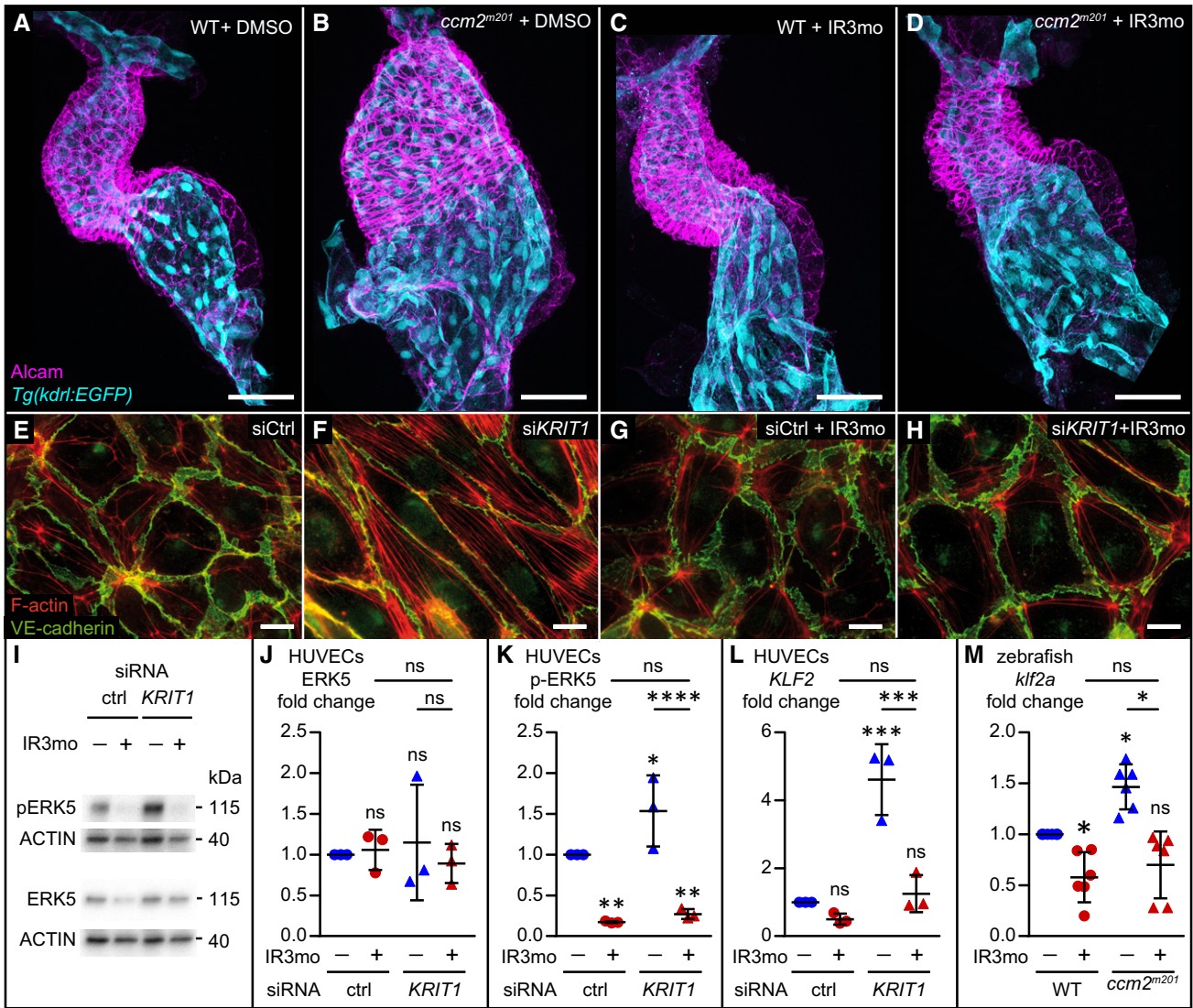

**Figure 4.  Indirubin-3-monoxime (IR3mo) rescues the CCM phenotype in zebrafish and HUVEC models.**

A–D   Treatment with 5 µM indirubin-3-monoxime (IR3mo) rescues the embryonic zebrafish *ccm2^{m201}* mutant ballooning heart phenotype. Shown are images of confocal z-scan projections of wild-type (WT) (A, C) and *ccm2^{m201}* mutant (B, D) zebrafish embryonic hearts at 48 hpf expressing the endothelial reporter *Tg(kdrl:GFP)^{s843}* (cyan) and counter-stained for Alcam (magenta, labeling the myocardium). Scale bar is 50 µm.

E–H   Treatment with 10 µM IR3mo for 48 h restores the wild-type ACTIN-adhesive phenotype in *KRIT1*-depleted HUVECs. Shown are confocal images of HUVECs with labeled VE-cadherin (green) and F-ACTIN (red). Control siRNA-silenced (E, G) and *KRIT1* siRNA-silenced (F, H) HUVECs were not treated (E, F) or treated (G, H) with IR3mo. Scale bar is 10 µm.

I–K   Treatment with IR3mo reduces phosphorylation of ERK (pERK) without affecting overall ERK protein levels. Shown in (I) are representative Western blots of *KRIT1*-silenced HUVECs lysates treated or not with 10 µM of IR3mo during transfection. ERK5 (J) and phosphorylated ERK5 protein levels (K) were measured relative to ACTIN protein levels based on three biological replicates with two technical replicates each (*n* = 3). Statistical analyses were performed using two-way ANOVA followed by Tukey's multiple comparisons test; error bars are SD. ****$P < 0.0001$; ns, not significant.

L   Treatment with IR3mo rescues the elevated *KLF2* levels of *KRIT1*-silenced HUVECs. RT-qPCRs were performed to measure *KLF2* levels, on three biological replicates (*n* = 3). Statistical analyses were performed using one-way ANOVA followed by Tukey's multiple comparisons test; error bars are SD. ***$P < 0.001$; ns, not significant.

M   Treatment with IR3mo rescues the elevated *klf2a* levels of *ccm2^{m201}* mutant zebrafish embryos at 48 hpf. RT-qPCRs were performed to measure *klf2a* levels, on six biological replicates (*n* = 6). Statistical analyses were performed using one-way ANOVA followed by Tukey's multiple comparisons test; error bars are SD. *$P < 0.05$; ns, not significant.

Source data are available online for this figure.

massive vascular lesions within the cerebellum by P8 (Fig 5A–D). We found that feeding pups with IR3mo alleviated the burden of lesions in both genetic conditions (Fig 5B and D). In particular, we

observed a significant reduction in the number of small lesions (< 5,000 mm²) under this treatment plan in both genetic models (Fig 5G and H). As a control, we found that treatment with IR3mo

had no effect on the width of the cerebellum (Fig 5E) and on the animal weight (Fig 5F) of the *iCCM2* mice at P8. These results indicate that IR3mo efficiently suppresses CCM phenotypes in zebrafish, mouse, and human endothelial cells.

## Discussion

Here, we describe a repurposing drug screen applied to zebrafish and *C. elegans* CCM models in which we uncovered 452 compounds that exhibited suppressive activity in at least one of the organisms. A loss of CCM proteins affects the formation or function of tube-like structures from the *C. elegans* excretory canal to the zebrafish heart tube. Our approach to use pharmacological suppression to restore the tissue architecture that is disturbed in CCM mutants involved organismal and cell-based assays. This provided a highly integrated platform that exceeded the complexity of previous purely cell-based high-throughput screens (Gibson *et al*, 2015; Nishimura *et al*, 2017) and, hence, provided a more comprehensive overview of relevant pathways

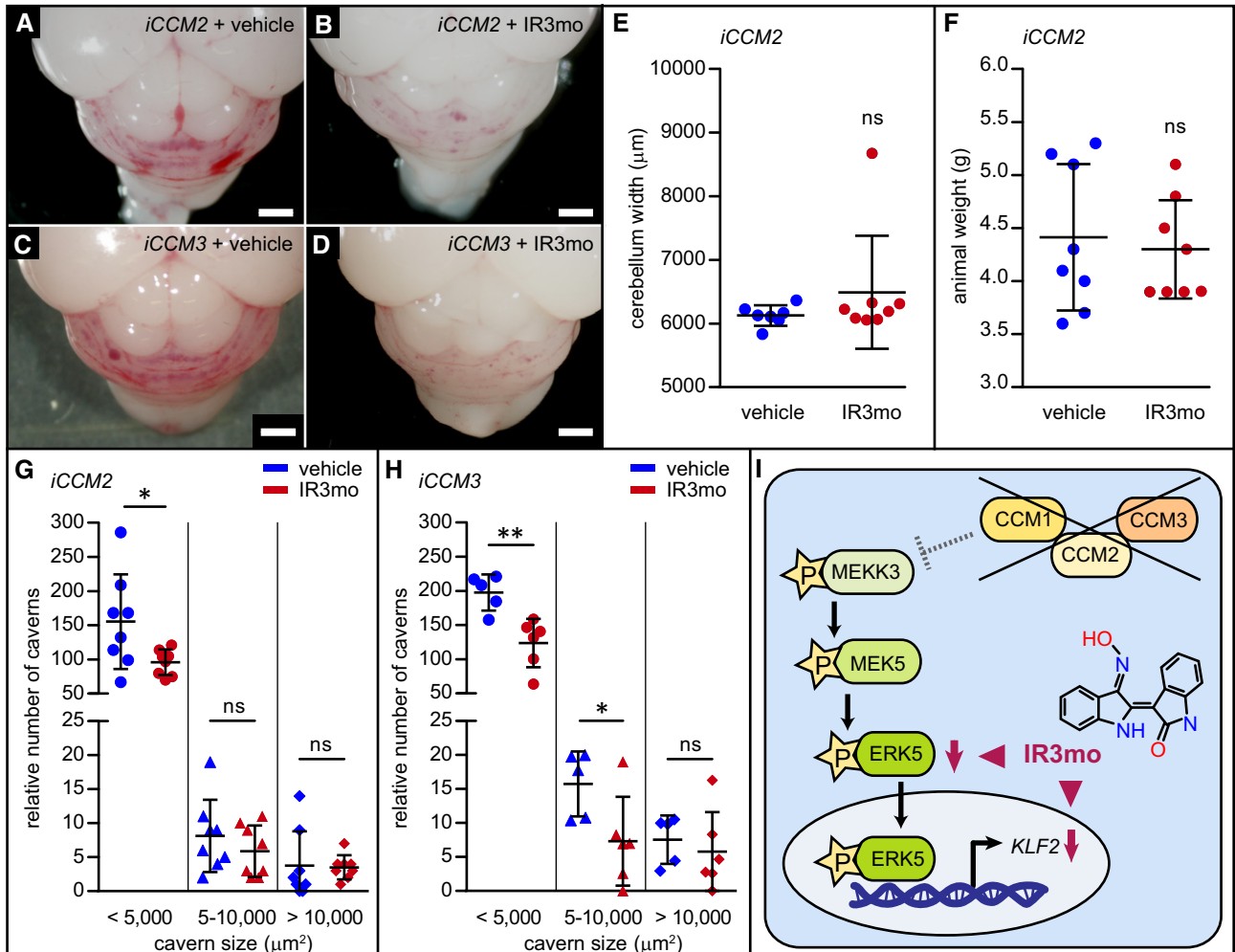

**Figure 5.    Treatment of *iCCM2* and *iCCM3* mice with indirubin-3-monoxime (IR3mo) alleviates the lesion burden within the cerebellum.**

A–D    Shown are representative pictures of brains at P8 from *iCCM2* (A, B) or *iCCM3* (C, D) mice treated with vehicle (A, C) or with IR3mo (B, D). Mice were treated daily between P2 and P7 with vehicle or with IR3mo. Scale bar is 1 mm.

E    Treatment with IR3mo does not affect cerebellum width at P8 of *iCCM2* mice. The measurements were performed on seven vehicle-treated and eight IR3mo-treated mice ($n = 7$, $n = 8$). Student's two-tailed $t$-tests were performed; error bars are SD. ns, not significant.

F    Treatment with IR3mo does not affect animal weight at P8 of *iCCM2* mice. The measurements were performed on eight vehicle-treated and eight IR3mo-treated mice ($n = 8$, $n = 8$). Student's two-tailed $t$-tests were performed; error bars are SD. ns, not significant.

G, H    Quantifications of relative numbers of caverns within three categories based on cavern size. Cerebellar regions were measured in *iCCM2* and *iCCM3* mutants at P8 after treatment with vehicle ($n = 8$ for *iCCM2* and $n = 5$ for *iCCM3*) or IR3mo ($n = 8$ for *iCCM2* and $n = 6$ for *iCCM3*). Student's two-tailed $t$-tests were performed; error bars are SD. ns, not significant; *$P < 0.05$; **$P < 0.01$.

I    Model scheme showing the molecular effect of IR3mo treatment on the ERK5 phosphorylation levels and the *KLF2* expression levels in the CCM loss-of-function context. Two arrowheads indicate different molecular effects of IR3mo within the ERK5-KLF2 pathway.

affected in the disease. While these screens identified a number of compounds that had not previously been implicated in CCM, little overlap with other screens or known suppressing compounds was reported. As a proof of the efficacy of our drug screen, we identified many of the compounds or signaling pathways that had been discovered in previous screens or molecular studies (Table EV1). We used the target prediction tool DePick (Liu *et al*, 2016) to identify known targets including the innate immunity receptor TLR4 (Tang *et al*, 2017), the metalloproteinases (MMP2, MMP7, MMP13, and MMP14; Zhou *et al*, 2015), and HMGCR (Nishimura *et al*, 2017). In addition, we found a number of compounds, both known and new, which affected molecular processes or physiological functions that appear to be relevant for CCM. These included anti-oxidants, analgesics, neurotransmitter and ion channel modulators, protein kinase inhibitors, and drugs with anti-hypertensive, anti-angiogenic, anti-inflammatory/immunosuppressive, or anti-mitotic effects (Figs EV2 and EV3). We were surprised to find that approximately 7–8% of all the compounds we tested had some effect on the phenotype in zebrafish or *C. elegans*. This could be due to the pleiotropy of molecular pathways that are relevant for CCM and to the enrichment of bioactive molecules within the compound libraries that were used. Our results strongly support the initial hypothesis that such unbiased screens can identify compounds with beneficial effects in the treatment or prevention of CCMs. The inventory of relevant compounds also raises the possibility of benefits to be gained through combinatorial treatments based on drugs which target different molecular pathways involved in the disease.

While whole-transcriptome analyses of CCM mutant animal models have led to the identification of a number of misregulated molecular pathways, clarifying their relevance to the disease has remained a challenge (Renz *et al*, 2015). By employing small molecules for probing relevant pathways and integrating transcriptomics data with systems pharmacology analyses, we have now identified a CCM-related signature that may point at critical hubs including the VEGFR signaling pathway which will be important for therapeutic applications. Unexpectedly, this analysis has revealed a striking conservation of the effects of drugs on worms and vertebrates, suggesting that some of these compounds may also have beneficial effects in translational therapeutic applications in the human disease. For instance, some of the compounds identified in both *C. elegans* and zebrafish were anti-hypertensive or anti-angiogenic drugs. Of note, worms have no circulatory system, and hence, these drugs may act by affecting more fundamental functions of conserved molecular pathways—including roles that go beyond those that have been well-established in vertebrate physiology. Similarly, neurotransmitter-related agonists or antagonists had beneficial properties in both worm and zebrafish. This class of drugs may function either autonomously on non-neural tissues or non-cell autonomously on neural cells in their interaction with the vasculature. Several compounds that affect MAPK signaling exhibited suppressive effects on CCM mutant phenotypes in worms and zebrafish. This finding suggests some cellular role of MAPK signaling in both apoptosis in the worm and roles in the vertebrate vasculature; until now, apoptosis was not among the functional GO-BP terms for the *ccm2* mutant transcriptome. Finally, metabolites of the retinoic acid synthesis pathway and

drugs that affected the metabolic enzymes of this biochemical synthesis pathway showed a potent capacity to suppress the CCM phenotypes in zebrafish and *C. elegans*. Further studies are required to assess the molecular role of this pathway upon the loss of CCM proteins.

The effects of some small molecules may be due to targeting multiple relevant pathways. An example is IR3mo, which is a derivative of indirubin, the bioactive ingredient in a traditional Chinese herbal medicine (Eisenbrand *et al*, 2004; Williams *et al*, 2011). Besides their anti-carcinogenic, anti-mitotic, and anti-inflammatory effects, indirubin and its derivatives interfere with angiogenic growth in the zebrafish embryo (Tran *et al*, 2007) and in HUVECs. This may be due to an inhibition of VEGFR2 signaling (Zhang *et al*, 2011). Other known targets of the drug include CDKs (Hoessel *et al*, 1999; Davies *et al*, 2001) and GSK3β (Leclerc *et al*, 2001). Indirubin compounds also affect signaling by Notch 1 (Lee *et al*, 2008), β1 integrin (Kim *et al*, 2011), β-catenin (Zahoor *et al*, 2014), and TGFβ/BMP (Cheng *et al*, 2012), and inhibit lipopolysaccharide-induced inflammation via TLR4, which is mediated by the NF-κB and MAPK signaling pathways (Lai *et al*, 2017). In mice, the lipopolysaccharide-induced stimulation of TLR4 causes acute lung injury due to an increase in oxidative stress and inflammation; indirubin treatment alleviates this effect (Qi *et al*, 2017). Here, we found that IR3mo reversed the molecular effects of activated MAPK signaling, which causes the enhanced phosphorylation of ERK5 in *siCCM1/KRIT1*-treated HUVECs and an increase in the transcriptional expression of *klf2a* mRNA in zebrafish *ccm2^{m201}* mutants (Renz *et al*, 2015). This shows that treatment with IR3mo not only reduces the burden of lesions in murine preclinical models of *CCM2* and *CCM3*, but also has an impact on a key molecular pathway involving MAPK signaling and endothelial *KLF2* activation that is functionally critical for *ccm* mutant cardiovascular defects in zebrafish and mouse (Renz *et al*, 2015; Zhou *et al*, 2015, 2016; Fig 5I). Although the precise mechanisms underlying disease etiology in CCM mouse models are still unknown, probing the effects of IR3mo will surely provide additional hints as to the molecular mechanisms involved in the pathology. Interestingly, DePick data have been an important tool not only for target protein prediction but, equally important, for excluding less likely target proteins. While GSK3β or CDK1/2 is well-characterized targets of IR3mo, none of them is significantly targeted by other compounds identified in the zebrafish screen (Table EV6). Hence, systems biological approaches of identifying positive and ruling out negative target proteins will further help to clarify CCM pathology-relevant regulatory networks. Currently, we can only speculate why IR3mo was not effective in *C. elegans*. For instance, the worm cuticle may not be permeable for the compound or IR3mo may target some of the pathways that are specific for vertebrates.

In conclusion, this study reveals a number of novel compounds affecting several molecular pathways and biological processes that are conserved and relevant to both the basic cellular and more complex cardiovascular defects found in CCM animal models relevant to the CCM pathology in humans. In the future, systems biological tools will help to improve our understanding of how these molecular pathways are interconnected, which will greatly aid in designing effective targeting therapeutics.

# Materials and Methods

### Screen database

The compound collections [GlaxoSmithKline Published Kinase Inhibitor Set (PKI, William Zuercher), The Library of Pharmacologically Active Compounds (LOPAC, Sigma-Aldrich), Spectrum (MicroSource Discovery Systems, Inc), Selleck Library] were assembled into a single structural data file, and unique identifiers were assigned using Konstanz Information Miner software (KNIME; Berthold *et al*, 2007). To check for structural overlap, we used the KNIME CDK extension module for normalizing the compound structures (desalting step) and calculation of extended connectivity fingerprints (ECFP-4; Beisken *et al*, 2013). Duplicate structures were subsequently identified by Tanimoto similarity score calculation. All screening results were integrated based on unique compound identifiers. Screens were analyzed blindly.

### *Caenorhabditis elegans* pharmacological suppression screen

Pharmacological suppression was adapted from liquid-based screening protocols (Lehner *et al*, 2006; Burns *et al*, 2015). Screens were carried out in 96-well plates using synchronized L1-stage *kri-1* (*ok1251*) mutant worms fed an *E. coli* HT115 bacterial strain expressing double-stranded RNA targeting *ccm-3* (*ccm-3* RNAi bacteria) from the Source Bioscience LifeSciences feeding library. RNAi knock-down was performed as previously described (Lehner *et al*, 2006) with several modifications. The RNAi bacteria were grown to mid-log phase and induced with isopropyl-β-D-thiogalactopyranoside (IPTG) at a final concentration of 0.4 mM for 1 h at 37°C with shaking, then pelleted, and concentrated 10-fold with liquid Nematode Growth Media supplemented with 25 μg/ml carbenicillin and 2.4 mM IPTG before dispensing into each well of the assay plates. The *kri-1* (*ok1251*) strain (Ito *et al*, 2010) was obtained from the Caenorhabditis Genetics Center. The *E. coli* HT115 strain carrying the L4440 empty vector was used as a control for RNAi machinery induction, and *pop-1* RNAi which produces a severe embryonic lethality phenotype was used as a control for RNAi induction efficiency.

Library compounds were screened with a concentration of 60 μM with a final concentration of DMSO of 0.6% v/v. Approximately 20 synchronized L1 larval stage *kri-1*(*ok1251*) worms were added to each well of the 96-well plates in 10 μl of M9 buffer. Plates were sealed with Parafilm and incubated for 13 days at 20°C with shaking at 200 rpm. The screening was performed using a dissection microscope by counting the number of living worms. The degree of pharmacological suppression of the *kri-1*(*ok1251*); *ccm-3* (RNAi) semi-lethal phenotype was assessed based on a comparison of viability in each compound to DMSO controls. The primary screen was carried out in duplicate and candidate suppressor compounds were retested at least once in triplicate using the same approach described above. Representative pictures of the *C. elegans* screen as shown in Fig EV1 were recorded in brightfield on a Leica MZ FLIII microscope with Leica 0.5× and 1.0× objectives and a Leica EC4 camera.

### Zebrafish handling and maintenance

Handling of zebrafish was done in compliance with German and Brandenburg state law and monitored by the local authority for animal protection (LAVG, Brandenburg, Germany). The following strains of zebrafish were maintained under standard conditions as previously described (Westerfield *et al*, 1997): *ccm2^m201^* and *krit1^ty219c^* (Mably *et al*, 2006); *Tg*(*kdrl:GFP*)^s843^ (Jin *et al*, 2005); and *Tg*(*myl7:EGFP*)^twu34^ (Huang *et al*, 2003).

### Zebrafish pharmacological suppression screen

The zebrafish screen was performed by incubating approximately 15-20 embryos per well in 24-well plates. Screening plates were prepared by automatically pipetting a 5 μl compound solution from the LOPAC, Selleck, or Spectrum stock libraries (1 mM stocks, in DMSO) into each well. Negative controls were wells without compounds. At 17-19 hpf, dechorionated zebrafish embryos were transferred within a volume of 0.5 ml E3 medium (5 mM NaCl, 0.17 mM KCl, 0.33 mM CaCl$_2$, and 0.33 mM MgSO$_4$) into each well and incubated for 24 h at 28.5°C. For indirubin-3-monoxime treatment (IR3mo, Sigma, #I0404), zebrafish embryos were incubated in 5 μM IR3mo/0.1% DMSO/E3 between 16 and 48 hpf. Control embryos were treated with 0.1% DMSO/E3.

Live visual inspection was conducted under a fluorescence stereomicroscope. The cardiac phenotype of *ccm2^m201^* mutants was assessed for rescue with the wild-type siblings serving as internal controls in each well. A pharmacological effect was considered a rescue when *ccm2^m201^* mutant hearts had a wild-type appearance or had a less severe phenotype even when wild-type embryos were also affected (e.g., the presence of blood flow, smaller heart, smaller inflow tract size, and constricted atrioventricular canal). Candidate suppressor compounds were re-screened. Representative pictures, as shown in Fig 1 of heads of embryos fixed with 4% paraformaldehyde (PFA) and embedded in 1% low-melting agarose, were recorded at a LSM710 confocal microscope (Zeiss) with a 10× objective.

### Zebrafish whole-mount immunohistochemistry

Zebrafish whole-mount immunohistochemistry, as shown in Figs 4 and EV5, was performed on 48 hpf embryos as previously described (Renz *et al*, 2015). The following antibodies were used: mouse Zn-8/Alcam (1:25, Developmental Studies Hybridoma Bank) and DyLight 649-conjugated goat anti-mouse (1:200, Jackson ImmunoResearch Laboratories, #115-495-003). Embryonic hearts were then prepared and mounted in SlowFade Gold (Invitrogen, #S36936). Images were recorded at a LSM 710 confocal microscope (Zeiss) with a 40× objective.

### Zebrafish RT-qPCR

Total RNA was extracted with TRIzol (Sigma) from pools of 7–15 embryos (48 hpf), and cDNA was synthesized with the RevertAid H Minus First Strand cDNA Synthesis kit (Thermo Fisher Scientific). Quantitative real-time PCRs (RT-qPCR) were performed as previously described (Renz *et al*, 2015) in compliance with the MIQE standard (Bustin *et al*, 2009) with six biological replicates, using 6 ng cDNA per reaction. Measurements were performed on a PikoReal 96 Real-Time PCR System (Thermo Fisher Scientific). Ct values were determined by PikoReal software 2.2 (Thermo Fisher Scientific). Results were analyzed using the comparative threshold cycle (Ct) method (2-ΔΔCt) to compare gene expression levels between samples as previously

described (Renz *et al*, 2015). As an internal reference, we used zebra-fish *eif1b* (Renz *et al*, 2015). The following primers were used:

| Primer name | Primer sequence | UniGene identifier |
|---|---|---|
| Klf2aDr_fwd | 5′-CTGGGAGAACAGGTGGAAGGA-3′ | Dr.29173 |
| Klf2aDr_rev | 5′-CCAGTATAAACTCCAGATCCAGG-3′ | |
| Eif1bDr_fwd | 5′-CAGAACCTCCAGTCCTTTGATC-3′ | Dr.162048 |
| Eif1bDr_rev | 5′-GCAGGCAAATTTCTTTTTGAAGGC-3′ | |

Since each single biological replicate represents an independent experiment from an independent clutch of fish, one-way ANOVA with Tukey's multiple comparisons test was performed.

WT + DMSO: 1 (SD: 0); *ccm2* + DMSO: 1.466 (SD: 0.22); WT + IR3mo: 0.5778 (SD: 0.2458); *ccm2* + IR3mo: 0.6992 (SD: 0.3278); $n = 6$ experiments.

| Tukey's multiple comparisons test | Adjusted *P*-value |
|---|---|
| wt + DMSO versus *ccm2* + DMSO | 0.013 (*) |
| wt + DMSO versus wt + IR3mo | 0.0305 (*) |
| wt + DMSO versus *ccm2* + IR3mo | 0.2296 (ns) |
| *ccm2* + DMSO versus wt + IR3mo | < 0.0001 (****) |
| *ccm2* + DMSO versus *ccm2* + IR3mo | 0.0144 (*) |
| wt + IR3mo versus *ccm2* + IR3mo | 0.843 (ns) |

## HUVECs pharmacological suppression screen

Human umbilical vein endothelial cells (HUVECs) from pooled donors (Lonza, #C2519A) were grown in EBM-2 basal medium (Lonza, #CC-3156) enriched with EGM-2 (Lonza, #CC-4176) growth factors at 37°C and 5% $CO_2$ in a humidified cell chamber. They were infected with pSICOR lentivirus expressing CCM2 shRNA 5′ CACACTGTGGTGTTGTCATTG 3′. This led to 70% reduction in CCM2 expression, as verified by RT-qPCR.

| shRNA | ctrl | shCCM2 |
|---|---|---|
| ATP50 | 1 | 0.34184 |
| Ranbp1 | 1 | 0.38087 |
| RPLO | 1 | 0.20438 |
| mean | 1.0 | 0.309 |
| SEM | 0.0 | 0.05352 |

Cells were then grown in a 384-well plate containing a different 10 μM compound solution in each well; negative controls were wells containing 0.2% DMSO or no additive. Cells were then fixed with 4% PFA and stained for the ACTIN cytoskeleton with Alexa Fluor 647 Phalloidin to assess whether the cell elongation and stress fiber phenotypes characteristic for CCM loss of function were rescued. This experiment was performed in triplicate, as well as repeated using compounds at 5 μM. Compounds were obtained from the Sigma, Spectrum, and Selleck Libraries. The rescue efficacy of the compounds was assessed blindly.

## Indirubin-3′-monoxime treatment of HUVECs

HUVECs from pooled donors (Lonza, #C2519A) were grown in EBM-2 basal medium (Lonza, #CC-3156) enriched with EGM-2 (Lonza, #CC-4176) growth factors at 37°C and 5% $CO_2$ in a humidified cell chamber. Cells were transfected twice at a 24-h interval with 30 nM *KRIT1* siRNA oligonucleotides (Ambion, #146529), *CCM2* siRNA (Dharmacon #L-014728-01), si*CCM3* (Dharmacon #L-004436-01), or CT siRNA (Dharmacon, #D-001810-01) in Optimem (Gibco, #31985-047) using RNAi Max lipofectamine (Invitrogen, #13778-150) as described in manufacturer's instructions. HUVECs were treated with 10 μM IR3mo (Sigma, #I0404) during the time of transfection. Untreated cells were used as controls.

## Immunohistochemistry of indirubin-3′-monoxime-treated HUVECs

HUVECs were grown and transfected as above, then seeded at confluency (300,000 cells/well) on cover glasses in a 24-well plate coated with fibronectin at 6 μg/cm$^2$ (Faurobert *et al*, 2013). The cells were treated with 10 μM IR3mo for 48 h prior to fixation with 4% PFA for 15 min at 37°C. Coverslips were blocked with 10% goat serum and incubated with the primary antibody VE-Cadherin mouse (Millipore, #MABT129) (1:200) for 1 h at 37°C and then with the Goat anti-Mouse IgG Secondary Antibody Alexa Fluor 488 (Thermo Fisher Scientific, #A-11029) and TRITC-Phalloidin (Sigma, #P-1951). Coverslips were then mounted in Mowiol. Immunofluorescence pictures, as shown in Figs 4 and EV5, were acquired at an Axioimager microscope (Zeiss) with an Orca R2 N/B camera (Hamamatsu).

## Western blotting of indirubin-3′-monoxime-treated HUVECs

Cells were lysed in Laemmli buffer on ice. Lysates were sonicated and boiled, and proteins were separated by SDS–PAGE using 10% acrylamide gel and transferred on a PVDF membrane. Membranes were blocked 45 min at room temperature with a 5% BSA/TBST solution. The primary antibody was incubated overnight at 4°C and HRP secondary antibody during 1 h at room temperature. Signal was detected using Clarity™ Western ECL Substrate (Bio-Rad) and Bio-Rad Chemidoc imaging system. Western blots are representative of three experiments. Signals were quantified using Image Lab (Bio-Rad). The following antibodies were used for immunoblots: phospho-ERK5 (Cell Signaling, #3371), ERK5 (Cell Signaling, #3372), ACTIN (Sigma, #A 3853), Peroxidase AffiniPure Goat Anti-Mouse IgG (Jackson Immuno Research, #115-035-174), and Peroxidase AffiniPure Fragment Donkey Anti-Rabbit IgG (Jackson Immuno Research, #711-036-152).

ERK5 and pERK5 protein levels were measured relative to ACTIN protein levels in a series of three biological replicates with two technical replicates each. Two-way ANOVA analysis was then performed followed by Tukey's multiple comparisons test.

Relative ERK levels: untreated control: 1 (SD: 0); control + IR3mo: 1.06 (SD: 0.2463); untreated KRIT1: 1.15 (SD: 0.709); KRIT1 + IR3mo: 0.8926 (SD: 0.2403). Relative pERK levels: untreated control: 1 (SD: 0); control + IR3mo: 0.1721 (SD: 0.0117);

untreated KRIT1: 1.537 (SD: 0.4362); KRIT1 + IR3mo: 0.2709 (SD: 0.061).

| | Tukey's test on ERK5/ACTIN; adjusted P-value | Tukey's test on pERK5/ACTIN; adjusted P-value |
|---|---|---|
| ctrl versus ctrl + IR3 | 0.959 (ns) | 0.0034 (**) |
| ctrl versus siKRIT1 | 0.6151 (ns) | 0.0278 (*) |
| ctrl versus siKRIT1 + IR3 | 0.8117 (ns) | 0.0083 (**) |
| ctrl + IR3 versus siKRIT1 | 0.8765 (ns) | < 0.0001 (****) |
| ctrl + IR3 versus siKRIT1 + IR3 | 0.5337 (ns) | 0.9436 (ns) |
| siKRIT1 versus siKRIT1 + IR3 | 0.1996 (ns) | < 0.0001 (****) |

## RT-qPCR of indirubin-3′-monoxime-treated HUVECs

Quantitative real-time PCR was performed in compliance with MIQE guidelines (Bustin et al, 2009) with iTaqTM Universal SYBR Green Supermix (Bio-Rad) on a C-1000 Touch Thermal Cycler (Bio-Rad) as previously described (Renz et al, 2015). Ct values were determined with the Bio-Rad CFX Manager. The following primers were used for RT-qPCR:

| Primer name | Primer sequence | UniGene identifier |
|---|---|---|
| For_KLF2 | 5′-CATCTGAAGGCGCATCTG-3′ | Hs.744182 |
| Rev_KLF2 | 5′-CGTGTGCTTTCGGTAGTGG-3′ | |
| For_ATP5O | 5′-ATTGAAGGTCGCTATGCCACAG-3′ | Hs.409140 |
| Rev_ATP5O | 5′-AACAGAAGCAGCCACTTTGGG-3′ | |
| For_RPLP0 | 5′-TGCTCAACATCTCCCCCTTCTC-3′ | Hs.546285 |
| Rev_RPLP0 | 5′-ACTGGCAACATTGCGGACAC-3′ | |
| For_RANBP2 | 5′-TGTAGTGATACTGATGAAGACAATGG-3′ | Hs.199561 |
| Rev_RANBP2 | 5′-TTGTGCTAGTTATTTCTTCTGTCTGAG-3′ | |

Results were analyzed using the comparative threshold cycle (Ct) method (2-ΔΔCt) to compare gene expression levels between samples as previously described (Livak & Schmittgen, 2001). The RT-qPCR experiment was performed three times with three technical replicates per experiment; KLF2 levels were normalized against three reference genes RANBP2, ATP50, and RPLO. One-way ANOVA analysis was then performed and followed by Tukey's multiple comparisons test.

Untreated control: 1 (SD: 0); untreated KRIT1: 4.613 (SD: 1.044); control + IR3mo: 0.4997 (SD: 0.1626); KRIT1 + IR3mo: 1.251 (SD: 0.5463).

| Tukey's multiple comparisons test | Adjusted P-value |
|---|---|
| ctrl versus ctrl + IR3 | 0.7376 (ns) |
| ctrl versus siKRIT1 | 0.0003 (***) |
| ctrl versus siKRIT1 + IR3mo | 0.9526 (ns) |
| ctrl+IR3 versus siKRIT1 | 0.0001 (***) |
| ctrl+IR3 versus siKRIT1 + IR3mo | 0.4558 (ns) |
| siKRIT1 versus siKRIT1 + IR3mo | 0.0006 (***) |

## Indirubin-3′-monoxime treatment of iCCM2/3 mice

The iCCM2 (CCM2 flox/flox; Cdh5(PAC)-CreERT2) and iCCM3 (CCM3 flox/flox; Cdh5(PAC)-CreERT2) mouse lines were previously reported (Boulday et al, 2011; Maddaluno et al, 2013) and maintained on a C57BL/6 background. Analyses of littermate pups were performed at P7 or P8 (as indicated) without gender consideration. Tamoxifen (Sigma) induction was performed as previously described (Boulday et al, 2011) using a single intra-gastric injection (20 μg/g). IR3mo (Sigma, cat# I0404) was administrated daily from P2 to P7 at the oral dose of 10 mg/kg (diluted in 2% DMSO-sunflower oil). Vehicle (diluent only) was administered following the same protocol to obtain the control non-treated group. All experimental animal procedures and mouse handling described in this study were in full accordance with the European directive regarding the protection of animals used for scientific purposes (Directive 2010/63/UE) and obtained authorization from the French Ministry of Research after approval from the "Lariboisiere-Villemin" Ethic Committee on animal testing (APAFIS#2769-201511061228356v3). Animals were housed and bred in our local conventional animal facility using standard individually ventilated cages, 12:12 light/dark cycles, and ad libitum access to food and water.

For the iCCM2 experiment, eight IR3mo-treated and eight vehicle-treated iCCM2 animals from three different litters were used. For the iCCM3 experiment, six IR3mo-treated and five vehicle-treated iCCM3 animals from two different litters were used.

We first assessed the weight of iCCM2 mice fed with vehicle (4.4 ± 0.24 g; n = 8) or with IR3mo (4.3 ± 0.16 g; n = 8) and their cerebellum width (vehicle: 6126.3 ± 60.90 μm; n = 7; IR3mo: 6491.6 ± 314 μm; n = 8). Student's two-tailed t-tests showed that there was no statistical significance between the two groups.

## Assessment of lesion burden in iCCM2/3 mice

Lesion burden quantification was performed using digital imaging resources (motorized Eclipse 80i, Nikon, combined to a high-sensitivity cmos sensor camera) and a specialized image analysis software (Nis Element AR, Nikon). Serial paraffin sections of half cerebellum were prepared (10 μm thick), and hematoxylin and eosin (H&E) staining was performed on every 10 sections. Cerebral lesion burden was quantified on each mosaic-reconstituted image of the entire cerebellar section. Lesion were categorized depending of their area (< 5,000 μm$^2$; 5–10,000 μm$^2$; and > 10,000 μm$^2$). Results are expressed as relative to the mean of total cerebellar area analyzed. Student's two-tailed t-tests were used to determine statistical significance between two groups. The significance level was set at $P < 0.05$.

For iCCM2 (n = 8 for vehicle and IR3mo):

| Size of lesions | Tissue | Mean number of lesions | SD | t-test; P-value |
|---|---|---|---|---|
| < 5,000 μm$^2$ | Vehicle | 155.4 | 69.18 | 0.0338(*) |
| | IR3mo | 95.75 | 18.77 | |
| 5–10,000 μm$^2$ | Vehicle | 8.125 | 5.303 | 0.3457 (ns) |
| | IR3mo | 5.875 | 3.796 | |
| > 10,000 μm$^2$ | Vehicle | 3.75 | 5.064 | 0.8970 (ns) |
| | IR3mo | 3.5 | 1.773 | |

For *iCCM3* ($n$ = 5 for vehicle and $n$ = 6 for IR3mo):

| Size of lesions | Tissue | Mean number of lesions | SD | *t*-test; *P*-value |
|---|---|---|---|---|
| < 5,000 µm² | Vehicle | 197.7 | 26.34 | 0.0038 (**) |
| | IR3mo | 123.6 | 35.37 | |
| 5–10,000 µm² | Vehicle | 15.73 | 4.773 | 0.0403 (*) |
| | IR3mo | 7.305 | 6.525 | |
| > 10,000 µm² | Vehicle | 7.548 | 3.561 | 0.5692 (ns) |
| | IR3mo | 5.775 | 5.837 | |

## Image and data analyses

Images were processed using common Fiji (Schindelin *et al*, 2012) and Adobe Systems Bridge and Photoshop functions such as crop, rotation, and white balance. Schemes were drawn, and figures were assembled in Adobe Illustrator. Statistical data were analyzed using Prism (GraphPad).

## Medical subject headings (MeSH) categories of compounds

For MeSH category annotations of compounds, we used PubChem CIDs corresponding to the InChI codes of compounds using the pcp.get_compounds command of the publicly available python script PubChemPy. We mapped to PubChem CIDs 1,497 and 4,306 of the compounds tested in zebrafish and *C. elegans*, respectively. Afterward, we queried rentrez (Winter, 2017), the Entrez R package from NCBI, with those PubChem CIDs to obtain, when assigned, the associated MeSH tree numbers in the category D ("Chemicals and Drugs") and subcategory D27 ("Chemical Actions and Uses"). Finally, we classified the compounds into two subcategories of the following three branches of the MeSH subcategory D27: "Molecular Mechanisms of Pharmacological Action" [D27.505.519], "Physiological Effects of Drugs" [D27.505.696], and "Therapeutic Uses" [D27.505.954].

## DePick analysis of compounds

To determine pharmacological targets enriched in the set of compounds active in zebrafish and *C. elegans* screens, we applied an updated version of the published DePick target deconvolution approach (Liu *et al*, 2016). Briefly, we obtained the known and predicted targets (> 50% precision) of active as well as the inactive compound sets of each organismal chemical screen using HitPickV2 (see detailed description below). After discarding highly promiscuous compounds (with more than 100 known or predicted protein targets), we applied the hypergeometric test (Benjamini–Hochberg FDR correction < 10%) to search for pharmacological targets enriched in the set of active compounds from the zebrafish and *C. elegans* screens.

## HitPickV2

HitPickV2 is an advanced version of the HitPick target prediction method (Liu *et al*, 2013) that applies a novel ligand-based approach to predict of up to 10 protein targets (among 2,736 human proteins) per compound. For each query compound, HitPickV2 identifies the closest, structurally similar compounds in the chemical-protein interaction space (which consists of 891,629 annotated compound-target associations) using k-nearest neighbors (k-NN; Nidhi *et al*, 2006) chemical similarity search and selects the space covering 10 distinct protein targets. Then, HitPickV2 scores and assigns a precision to these predicted targets based on the performance of HitPickV2 in an in-house database assessed by cross-validation. The precision depends on the number of target occurrence within the restricted space of 10 targets, the ranking of the association of the compound with those targets based on the scores of Laplacian-modified naive Bayesian models (Rogers & Hahn, 2010), and the Tanimoto coefficient between the query and the most similar compound with the annotated predicted target in such space.

## Analysis of significant value for common shared active compounds between zebrafish and *C. elegans* screens

We ran permutation analyses to assess the random expectation of the number of active compounds shared by the *C. elegans* and zebrafish chemical screens. For that, we randomly shuffled 1,000,000 times the activity values of all compounds (1,600, 1,080 of them were tested in *C. elegans*) tested in the zebrafish screen and computed the number of active compounds also showing activity in the *C. elegans* screen (total number of compounds screened 4,748). Taking into account the distribution of the common active compounds in the random sets and those with a shared number of compounds equal or higher than 32 (actual number of shared common hits), we derive a *P*-value. The *P*-value obtained was 0, showing the statistical significance of the shared active compounds between the two screens.

## Gene ontology (GO) enrichment analysis of transcriptomics data and DePick targets

GO biological processes (GO-BP) enrichment analyses of differentially regulated genes of a zebrafish transcriptome experiment (Renz *et al*, 2015) as well as DePick *C. elegans* and zebrafish targets of chemical screens were performed using dedicated Perl scripts. Associations connecting GO-BP terms and differentially regulated genes (1,608) and background genes (10,652) in the transcriptome experiment were extracted from the gene_association.zfin goa_zebrafish.gpa files [downloaded from Gene Ontology Consortium (Ashburner *et al*, 2000; The Gene Ontology Consortium, 2017) on January 26, 2018)]. Associations between GO-BP terms and zebrafish (47) and *C. elegans* (134) significant human DePick targets as well as DePick background druggable proteins (2719 with a GO-BP term) from chemical screens were determined using the goa_human.gaf file (downloaded on January 30, 2018 from the Gene Ontology Consortium). We then propagated the GO-BP annotations to parent terms in the ontology linked via the "is a" relationship. We applied the hypergeometric test (Benjamini–Hochberg FDR correction < 10%) to determine GO-BP terms enriched in the set of differentially regulated genes in zebrafish transcriptome analysis (compared to the background genes) and the DePick significant targets in zebrafish as well as *C. elegans* chemical screens (compared to the druggable target background).

**The paper explained**

**Problem**

Cerebral cavernous malformations (CCM) are vascular lesions of the central nervous system vasculature for which a pharmacological cure is not yet available.

**Results**

In our study, we have performed pharmacological suppression screens using several well-established *CCM* animal models and identified novel active compounds. We have investigated the potential pathways targeted by these compounds in the context of CCM using bioinformatics. Additionally, we find that one of these active compounds, indirubin-3-monoxime, also alleviates the lesion burden in mouse models for *CCM*.

**Impact**

Indirubin-3-monoxime is a promising new compound in the treatment of CCM. Our screens have uncovered several novel CCM-relevant pathways with potential implications for our understanding of the CCM pathology.

**STRING**

Analyses were done with STRING (Search Tool for the Retrieval of Interacting Genes/Proteins) database version 10.5 (http://string-db.org; Szklarczyk *et al*, 2015); Basic settings (using all active interaction sources), minimum required interaction score: high confidence. The Human DePick targets derived from the zebrafish and from the *C. elegans* screens were analyzed against the DePick background (2,695 druggable targets that could be mapped to STRING identifiers). Network Statistics for zebrafish DePick: number of nodes: 47; number of edges: 75; average node degree: 3.19; avg. local clustering coefficient: 0.633; expected number of edges: 15; and PPI enrichment *P*-value: < 1.0e-16. Network Statistics for *C. elegans* DePick: number of nodes: 134; number of edges: 266; average node degree: 3.97; avg. local clustering coefficient: 0.492; expected number of edges: 99; and PPI enrichment *P*-value: < 1.0e-16. Edge Confidence: low (0.150); high (0.700); medium (0.400); highest (0.900).

**Expanded View** for this article is available online.

## Acknowledgements

We would like to thank M. Kneiseler for fish maintenance. For critical reading of the manuscript, we are indebted to R. Hodge and members of our team. We are also indebted to Minh Arnould for excellent technical help in CCM mouse model analysis. The entire screening consortium has been generously supported by the transnational E-RARE grant "CCMCURE". S.A.-S. was supported by the excellence cluster REBIRTH and SFB958. Support for W.B.D. came from E-RARE (ERL 138397) and the Canadian Institutes for Health Research (PJT 153000). E.F. and C.A.R. were supported by ARC LRCC and ANR.

## Author contributions

CO designed and performed the zebrafish experiments, analyzed the results, and wrote the manuscript with SA-S. KH and CIG contributed to the zebrafish experiments and to the analysis of results. GB designed mouse experiments, analyzed the results, and contributed to the manuscript redaction. CDL and CC performed *in vivo* experiments in mouse and quantified the lesion burden. ET-L directed the research project on *CCM* mouse models, contributed to scientific discussion, and provided critical comments to the manuscript. IV, SH, and MC performed computational analysis. MC provided a critical review of the manuscript. JK, MH, PR, and WBD performed the *C. elegans* suppression screen, analyzed and interpreted the results, and provided critical comments on the manuscript. ME, CA-R, and EF established the CCM-depleted HUVECs and helped with performing the compound screen, data analysis, and representation. Both CA-R and EF provided critical comments on the manuscript. MN, SR, and JPvK organized and distributed the small compound libraries for the zebrafish, worm, and HUVECs screens. They contributed to the analysis of CCM-depleted HUVECs. JPvK provided critical comments on the manuscript.

## Conflict of interest

The authors declare that they have no conflict of interest.

## For more information

(i) http://mips.helmholtz-muenchen.de/Depick/
(ii) USA association: http://www.angiomaalliance.org/
(iii) France association: http://association-cavernome-cerebral.e-monsite.com/
(iv) OMIM: https://www.omim.org
(v) *KRIT1 (CCM1)* gene number: 604214; *CCM2* gene number: 607929; *PDCD10 (CCM3)* gene number: 609118; CCM phenotype entry: 116860

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
