## [Review Process File · EMBO Molecular Medicine]

Systematic pharmacological screens uncover novel pathways involved in cerebral cavernous malformations

Cécile Otten, Jessica Knox, Gwénola Boulday, Mathias Eymery, Marta Haniszewski, Martin Neuschwander, Silke Radetzki, Ingo Vogt, Kristina Hähn, Coralie De Luca, Cécile Cardoso, Sabri Hamad, Carla Igual Gil, Peter Roy, Corinne Albiges-Rizo, Eva Faurobert, Jens P. von Kries, Mónica Campillos, Elisabeth Tournier-Lasserre, W. Brent Derry and Salim Abdelilah-Seyfried

Review timeline:

Submission date:	22 nd March 2018
Editorial Decision:	28 th March 2018
Author Appeal:	3 rd April 2018
Editorial Decision:	4 th April 2018
Revision received:	6 th April 2018
Editorial Decision:	3 rd May 2018
Revision received:	3 rd July 2018
Editorial Decision:	7 th August 2018
Revision received:	9 th August 2018
Accept:	13 th August 2018

Editor: Lise Roth

Transaction Report:

1st Editorial Decision

28th March 2018

Dear Prof. Seyfried,

Thank you for submitting your manuscript to EMBO Molecular Medicine. I have now had a chance to read your manuscript carefully and to discuss it with the other members of our editorial team. In addition, I have also sought external advice on the study from a good expert in the field. I am sorry to inform you that we find that the manuscript is not well suited for publication in EMBO Molecular Medicine and that we therefore have decided not to proceed with its handling and peer review.

Your study reports the results of a novel and integrative drug screen based on zebrafish, *C. elegans* and endothelial cell cultures models of cerebral cavernous malformations (CCM) that aimed at repurposing small compounds that would restore the wild-type phenotype in the 3 models. Computational tools further defined signaling pathways relevant to the disease and potential targets for small-molecule based therapies. Nevertheless, a different compound that is widely used in long-term clinical treatments for leukemia and other chronic diseases was selected based on its ability to interfere with several molecular pathways relevant to CCM: indirubin-3-monoxime (IR3mo). Feeding pups with IR3mo alleviated the burden of lesions in knock-out mouse models of CCM2 and CCM3, which was attributable to a reduced number of small lesions.

We recognize the interest and the technical solidity of your work. However, previous publication of similar screening and previous reports on the role of the identified pathways in CCM development detract from the kind of conceptual advance we expect from an EMBO Molecular Medicine article. Unfortunately, the expert external advisor we consulted with agreed with our concerns, which I am afraid preclude further consideration here.

Author Appeal

3rd April 2018

My colleagues and I very much appreciate your willingness and additional effort in reviewing our appeal.

Our manuscript summarizes the results of years of work of a large European-Canadian transnational consortium. The scope and results of our work exceed previous screens using CCM-deficient cell-based assays that were rather limited in their output and that fell short of providing a comprehensive overview of relevant pathways in CCM (Gibson et al., 2015; Nishimura et al., 2017). For instance, the screen by Nishimura et al. was highly biased and mainly focused on the mevalonate pathway which had already been implicated in CCM. Hence, the results of our study are urgently anticipated by the entire CCM community.

I would like to emphasize that what made the great difference of our screen was the completely unbiased, integrative, and multi-organism-based approach that had not previously been tried. We combined compound screens in the invertebrate model *C. elegans* and in the zebrafish embryo with its complex vertebrate cardiovascular system. Compounds that suppressed features and processes of CCM were then tested in CCM-deficient human endothelial cells and in preclinical mouse models of CCM1 and CCM3.

Our combined multi-organismal screens revealed a number of important insights into the regulatory network involved in CCM. This has come with many surprises that we discuss in our manuscript. For instance, we find a striking conservation of the effects of drugs on worms and vertebrates, suggesting that some of these compounds may also have beneficial effects in translational therapeutic applications in the human disease. Among the compounds identified in both *C. elegans* and zebrafish were anti-hypertensive or anti-angiogenic drugs. Of note, worms have no circulatory system and hence, these drugs may act by affecting more fundamental functions of conserved molecular pathways – including roles that go beyond those that have been well-established in vertebrate physiology.

Similarly, neurotransmitter-related agonists or antagonists had beneficial properties in both worm and zebrafish. This class of drugs may function either autonomously on non-neural tissues or non-cell autonomously on neural cells in their interaction with the vasculature. Several compounds that affect MAPK signaling exhibited suppressive effects on CCM mutant phenotypes in worms and zebrafish. This finding suggests some cellular role of MAPK signaling in both apoptosis in the worm and roles in the vertebrate vasculature; until now, apoptosis is not among the functional GO-BP terms for the *ccm2* mutant transcriptome.

Metabolites of the retinoic acid synthesis pathway and drugs that affected the metabolic enzymes of this biochemical synthesis pathway showed a potent capacity to suppress the CCM phenotypes in zebrafish and *C. elegans*. This is an exciting finding given a recent study by the team of Andreas Trumpp who showed that Vitamin A-retinoic acid signaling regulates hematopoietic stem cell dormancy (Cabezas-Wallscheid et al., 2017). Further studies are required to assess the molecular role of this pathway upon the loss of CCM proteins.

In conclusion, we identify both novel compounds and the molecular pathways and biological processes that they affect. These are conserved and relevant to both the basic cellular and more complex cardiovascular defects found in CCM animal models relevant to the pathology in humans. Our systems biological approach significantly improved our understanding of how these molecular pathways are interconnected, which will greatly aid in designing effective targeting therapeutics. The inventory of relevant compounds now raises the possibility of benefits to be gained through combinatorial treatments based on drugs which target different molecular pathways involved in the disease.

We hope that you will favorably consider our appeal. In addition to the other reviewer suggestions, we would also like to suggest Elisabetta Dejana as an excellent reviewer of our work.

2nd Editorial Decision

4th April 2018

Thank you for your e-mail asking us to reconsider our decision on your manuscript. I have now carefully read your letter and article again and discussed it with my colleagues, including our chief

editor.

I appreciate that you highlight once more the impact and novelty of the study. Reading your letter, I realized that indeed, some aspects of the work were not clear to us when we first evaluated your article:

1) For example, you mention that your screening strategy is original and novel compared to previous published screens, that these screens might have been biased and as such should not be considered as precedence. This type of information would be relevant for the community to add in the discussion section of the paper and backed up with scientific arguments. We are a general journal and our readership is broad, explanations and clarifications for a general audience are always welcome.

2) You also explain in the letter why the different pathways identified through the screen are particularly relevant in CCM. Our understanding now is that while the drug administered to mice was not identified through the screen, it targets some of the relevant pathways that were found relevant through the screen. Is that correct? If it is, I would strongly encourage you to rephrase this part of the work to make the reasons why you chose IR3mo more evident for the readers, or the study appears split with the screen on one side and a drug testing in mice on the other, with at best a weak link between the two. Moreover, IR3mo has no effect in *C. elegans*. It is unclear to us why that is. Could you please clarify in the text as well?

This said, after intense internal discussions about your paper, I would be happy to reconsider my decision and send your manuscript out for peer-review, provided you address these two points above to help with the understanding of the paper rationale.

I look forward to hearing from you soon. Should you decide to modify your paper accordingly and resubmit, please let us know.

1st Revision - authors' response

6th April 2018

Thank you very much for reconsidering our manuscript. We have read your excellent comments and the two major points are well-made. We understand that they are critical for a better understanding of this study and we will be happy to improve upon these points. I would also like to briefly reply to your questions below:

Best regards

Salim Seyfried

POINTS HIGHLIGHTED BY EDITOR.

1) For example, you mention that your screening strategy is original and novel compared to previous published screens, that these screens might have been biased and as such should not be considered as precedence. This type of information would be relevant for the community to add in the discussion section of the paper and backed up with scientific arguments. We are a general journal and our readership is broad, explanations and clarifications for a general audience are always welcome.

We will expand this discussion.

2) You also explain in the letter why the different pathways identified through the screen are particularly relevant in CCM. Our understanding now is that while the drug administered to mice was not identified through the screen, it targets some of the relevant pathways that were found relevant through the screen. Is that correct?

IR3mo has been identified during the screen (e.g. see tables S1,3). We realize that the reader may be surprised that it was not mentioned in the first part of the manuscript. We will strengthen the tie between the first and second part of the manuscript and better reveal the logic of analyzing IR3mo in preclinical trials (which is, among others and as discussed in the

first paragraph of the results chapter on IR3mo, that it targets some of the relevant pathways that were found relevant through the screen).

If it is, I would strongly encourage you to rephrase this part of the work to make the reasons why you chose IR3mo more evident for the readers, or the study appears split with the screen on one side and a drug testing in mice on the other, with at best a weak link between the two. Moreover, IR3mo has no effect in *C. elegans*. It is unclear to us why that is. Could you please clarify in the text as well?

We can only speculate why IR3mo is not working in worm. For instance, the worm cuticle may not be permeable for the drug or the concentration used in the trials may not have been sufficient. Another possibility is that IR3mo targeted some of the pathways that are specific for zebrafish (SRC, MST1R, VEGFR2). However, we decided that IR3mo was still an excellent candidate drug based on the logic as outlined above.

3rd Editorial Decision

3rd May 2018

Thank you for the submission of your manuscript to EMBO Molecular Medicine. We have now heard back from the three referees whom we asked to evaluate your manuscript.

As you will see from the reports below, while referees 2 and 3 are overall positive and support, in principle, publication of the article in EMBO Molecular Medicine (pending appropriate revisions), referee 1 questions the rationale for investigating IR3mo (as we did and previously discussed with you). Therefore, a more thorough discussion on the choice of IR3mo, as well as addressing the reviewers' concerns in full will be necessary for further considering the manuscript in our journal, with the exception of additional validation in other *in vivo* CCM models asked by referee 2 (referee 2, comments 2 and 4). EMBO Molecular Medicine encourages a single round of revision only and therefore, acceptance or rejection of the manuscript will depend on the completeness of your responses included in the next, final version of the manuscript.

REFeree REPORTS.

Referee #1 (Remarks for Author):

The manuscript attempts to find new small molecule compounds that can therapeutically alleviate the condition, cerebral cavernous malformations (CCM). They use two established CCM model systems in zebrafish, worm, and in huvec knockdown cells, to probe a panel of compounds for those that have effects in all three systems. Surprisingly, although they found 5 compounds that seemed to have effects in each system, they chose a kinase inhibitor that did not work in the worm for further studies. Using this inhibitor, in wt and siKRIT1 HUVECs they report reduced ERK5 phosphorylation, and in two mouse models of CCM disease they report rescue of lesion burden. However, they do not understand the mechanism of ERK5 phosphorylation reduction, and I think that understanding the mechanism of IR3mo inhibition of ERK5 phosphorylation is required for this study to be clinically informative. Some specific comments are noted below.

1. Their rationale to follow up Indirubin-3-monoxime, which rescued in only two of the models, and not one of the 5 compounds that rescued in all 3 is unclear. For example, one of these (ENMD-2076, which should be listed in the main text), is a kinase inhibitor (ENMD-2076) that also inhibits Aurora, KDR, FLT4, FLT3 and SRC. There seems to be much overlap with targets of indirubin-3-monoxime. Also, the protein interaction network (p12 referring to Table S8) discusses the importance of proteins involved in angiogenesis signaling, including FLT1, KDR and FLT4 which they didn't follow up experimentally (are the effects they see due to IR3mo inhibition of KDR or SRC tyrosine kinases, for example, and not MEK5 serine/threonine kinase as shown in Fig 4h, does IR3mo inhibit ROCK?). This leads one to wonder what the best candidate compound from their

screens was. Particularly as this study is trying to move towards a clinically-useful compound.

2. Figure 3CDE. The Western blots should be shown in the figure, and a scatter plot is the preferred method of data representation. The individual data points should be shown for each replicate. SD is preferred over SEM; Figure 5 of PMID 25204545 (Motulsky, 2014) illustrates why. Likewise Fig 4g should be shown as a scatter plot using SD not SEM.

3. The schematic in figure 4h is misleading: they do not know the mechanism of IR3mo modulation of ERK5 inhibition.

4. P9. Lines 3-4. They should include the full list of compounds shown in Table S2.

5. They should include line numbers in their manuscript.

Referee #2 (Remarks for Author):

Comments to the authors on the article titled,
"Systematic Pharmacological suppression uncovers novel molecular pathways involved in cerebral cavernous malformations"

This paper nicely reports the application of small-molecule suppression screens in different CCM-mutant models which revealed signaling pathways relevant to CCM diseases. The authors also identified indirubin-3-monoxime IR3mo compound as a prime candidate that alleviated the CCM-lesion burden in murine models and HUVECs. Altogether, the technicality of this study is robust and therefore, has the possibility to open up new platforms for future studies affecting several signaling pathways and biological processes relevant to CCM pathology.

However, there are a few concerns which need to be addressed:

Comment 1. Figure 3. Page 22-23: The authors applied STRING network clustering to visualize protein interaction networks related to CCM disease. This revealed several protein clusters both in the zebrafish and *C. elegans* model. However, implementing one network clustering is insufficient to determine the efficacy of the results obtained. In order to avoid biased results and achieve consistency, it would be relevant to apply multiple protein-protein interaction web based programs available online (such as: BioGRID, IntAct etc).

Comment 2. Figure 4a. The author used *ccm2* mutant as an example to show the effects of IR3mo on zebrafish embryos. It will worth to know whether IR3mo will have any changes on *ccm1* and *ccm3* mutants on zebrafish heart phenotype.

Comment 3. Figure 4b. Similarly it will be relevant to also see the phenotype of IR3mo on CCM2 or PDCD10 siRNA silenced HUVECs.

Comment 4. Figure 4f. The authors demonstrated the effect of treatment of IR3mo on *iCCM2* and *iCCM3* mice which alleviated the lesion burden. One might also try to look if there is any effect on lesion burden of IR3mo on *iKRIT1* mice model.

Referee #3 (Comments on Novelty/Model System for Author):

The use of multiple animal models for the initial screen, and then a cell culture model for validation of targets makes this a unique study and a very robust study design. The final mouse model, used for only one compound/drug, is important as well as the mouse model is the most relevant for future clinical trials. All in all a very nice and robust study with high scientific and potential medical impact, per my comments below.

Referee #3 (Remarks for Author):

This is a real tour de force in terms of effort and organization, and the study design is quite robust. The use of two different animal models for the drug screen makes this study unique, and the confirmatory cell culture model before moving to the more difficult but clinically relevant mouse model is another nice addition.

I am not an expert regarding the data mining and bioinformatic analyses. Since CCM pathobiology is already extensively studied, it was reassuring that quite a few drugs, drug targets, or pathways/biological processes were identified through the analysis that have already been suggested by other published studies. However, there seem to be so many different and apparently distinct pathways and biological processes that have resulted from the analyses that I would wonder how many are truly relevant to CCM pathobiology? When the analysis points in so many different directions, I have less confidence the relevance of each. However, one advantage to including all the bioinformatic analyses in this paper is that now there are new hypotheses to test concerning CCM pathobiology. So if I consider all this data analyses as hypothesis generating, I am more assured of its importance.

A final point is that IR3mo appears to be a very strong candidate for additional mechanistic studies, but also additional studies in animals that might lead to clinical trials. So the proof of the value of this study is that a new, exciting compound has been identified that might ultimately provide some relief to CCM patients. Congratulations to the investigative team.

2ND Revision - authors' response

3rd July 2018

Referee #1 (Remarks for Author):

The manuscript attempts to find new small molecule compounds that can therapeutically alleviate the condition, cerebral cavernous malformations (CCM). They use two established CCM model systems in zebrafish, worm, and in huvec knockdown cells, to probe a panel of compounds for those that have effects in all three systems. Surprisingly, although they found 5 compounds that seemed to have effects in each system, they chose a kinase inhibitor that did not work in the worm for further studies. Using this inhibitor, in wt and siKRIT1 HUVECs they report reduced ERK5 phosphorylation, and in two mouse models of CCM disease they report rescue of lesion burden. However, they do not understand the mechanism of ERK5 phosphorylation reduction, and I think that understanding the mechanism of IR3mo inhibition of ERK5 phosphorylation is required for this study to be clinically informative. Some specific comments are noted below.

1. Their rationale to follow up Indirubin-3-monoxime, which rescued in only two of the models, and not one of the 5 compounds that rescued in all 3 is unclear. For example, one of these (ENMD-2076, which should be listed in the main text), is a kinase inhibitor (ENMD-2076) that also inhibits Aurora, KDR, FLT4, FLT3 and SRC. There seems to be much overlap with targets of indirubin-3-monoxime. Also, the protein interaction network (p12 referring to Table S8) discusses the importance of proteins involved in angiogenesis signaling, including FLT1, KDR and FLT4 which they didn't follow up experimentally (are the effects they see due to IR3mo inhibition of KDR or SRC tyrosine kinases, for example, and not MEK5 serine/threonine kinase as shown in Fig 4h, does IR3mo inhibit ROCK?). This leads one to wonder what the best candidate compound from their screens was. Particularly as this study is trying to move towards a clinically-useful compound.

Upon completion of the primary screen in worms, zebrafish, and HUVECs, our consortium was facing the difficult decision to select not more than one or two compounds for preclinical trials in the murine *ccm* models. Given this limitation, we defined a number of criteria that served as a guideline for this decision. As indicated in our manuscript, one key aspect was the possibility of applying the drug safely for long-term usage without causing major side effects in patients. Furthermore, we ranked the positive effects of a drug in the zebrafish *ccm* model higher compared to the *C.elegans ccm* model since the zebrafish is a vertebrate with a cardiovascular system. As discussed in the text (lines 296-306), IR3mo is an FDA-approved drug with low toxicity derived from traditional Chinese medicine that has been widely used to treat leukemia and other chronic diseases (Eisenbrand et al, 2004; Williams et al, 2011) and interferes with several molecular pathways relevant to CCM. In comparison, treatment with other strong anti-angiogenic or anti-proliferative drugs (such as the multi-kinase inhibitor ENMD-2076) is associated with more severe side effects. Also, as shown in Table EV10 (former suppl. Table S10), IR3mo has at least 10 known

protein targets as predicted by DePick and other known targets including AHR, CDK5, CDK5R1, which are described DrugBank interaction partners according to PubChem. Potentially, IR3mo may affect a unique combination of targets that mediate the rescue effect.

As additional information, throughout the revisions phase, the Toronto teams of Brent Derry and Peter Roy applied different IR3mo treatment plans on *C. elegans ccm* models which did not yield a rescue effect. We conclude that IR3mo is highly effective in the context of vertebrate *ccm* models and human CCM-depleted endothelial cells but not in *C.elegans*.

2. Figure 3CDE. The Western blots should be shown in the figure, and a scatter plot is the preferred method of data representation. The individual data points should be shown for each replicate. SD is preferred over SEM; Figure 5 of PMID 25204545 (Motulsky, 2014) illustrates why. Likewise Fig 4g should be shown as a scatter plot using SD not SEM.

Within the revised manuscript, we have now improved old Figure 4 which has been divided into new Figure 4 (zebrafish and HUVEC data) and a new Figure 5 (mouse data and model): the Western blots (previous suppl. Figure S6) have been moved into new Figure 4 (Fig 4I). The representation of the quantifications shown in Figure 4C-E has been changed into scatter plots with error bars representing the SD (Fig 4J-M). Mouse data from previous Figure 4F-H and suppl. Fig S7 have been moved into new Figure 5A-H. We have also complied with EMBO Molecular Medicine standards and included the information about p-values, n sizes, and statistical tests within figure legends.

3. The schematic in figure 4h is misleading: they do not know the mechanism of IR3mo modulation of ERK5 inhibition.

This point has been addressed by modifying the model figure and the text of the figure legend to highlight the finding that IR3mo affects pERK5 protein levels and *KLF2* expression levels (which is a strong molecular readout of the CCM pathway). As indicated in that figure, two arrowheads are used to indicate these effects of IR3mo and we do not mean to imply that *KLF2* mRNA or ERK5 phosphorylation are directly affected by IR3mo. This is explicitly described within the revised figure legend.

4. P9. Lines 3-4. They should include the full list of compounds shown in Table S2.

We have changed the text accordingly and list the other compounds from Table EV2 that were missing in the main text (line 185-186). Similarly, we have also named the 5th compound that showed some degree of rescue in all three CCM models and that was missing from the main text (Table EV3) (lines 200-201).

5. They should include line numbers in their manuscript.

We have included line numbers within the revised manuscript.

Referee #2 (Remarks for Author):

Comments to the authors on the article titled,
"Systematic Pharmacological suppression uncovers novel molecular pathways involved in cerebral cavernous malformations"

This paper nicely reports the application of small-molecule suppression screens in different CCM-mutant models which revealed signaling pathways relevant to CCM diseases. The authors also identified indirubin-3-monoxime IR3mo compound as a prime candidate that alleviated the CCM-lesion burden in murine models and HUVECs. Altogether, the technicality of this study is robust and therefore, has the possibility to open up new platforms for future studies affecting several signaling pathways and biological processes relevant to CCM pathology.

However, there are a few concerns which need to be addressed:

Comment 1. Figure 3. Page 22-23: The authors applied STRING network clustering to visualize protein interaction networks related to CCM disease. This revealed several protein clusters both in

the zebrafish and *C. elegans* model. However, implementing one network clustering is insufficient to determine the efficacy of the results obtained. In order to avoid biased results and achieve consistency, it would be relevant to apply multiple protein-protein interaction web based programs available online (such as: BioGRID, IntAct etc).

The reviewer suggests to apply protein-protein interaction web based programs available online and suggests BioGRID and IntAct as examples of such programs. Unfortunately, BioGRID and IntAct are repositories of protein-protein interaction data but they do not provide online web programs for Network visualization. These (and other) resources storing protein-protein interaction information are instead commonly used by several web tools (programs) dedicated to protein Networks including STRING.

We chose STRING to visualize our Networks for its comprehensiveness –it uses the widest breadth of input sources, including automated text-mining and computational predictions - and more importantly, for its quality control - each interaction is annotated with benchmarked confidence scores-. The scores indicate the estimated likelihood that a given interaction is biologically meaningful, specific, and reproducible, given the supporting evidence (e. g experiments) (PMID: 27924014). We chose to display high confidence Networks by selecting only interactions highly supported (e.g. by more than one evidence) (high confident cut-off :0.7). With the choice of STRING, we also reduce and compensate the individual biases in information that each resource might have. While our Networks are not free of potential general bias in the currently existing protein-protein information data (e.g. certain proteins are better characterized than others and therefore, more interactions for them are known), they provide a more comprehensive and consistent picture of protein CCM Networks than those constructed with only one source of protein-protein interaction information (e.g. only BioGRID or IntAct).

Comment 2. Figure 4a. The author used *ccm2* mutant as an example to show the effects of IR3mo on zebrafish embryos. It will worth to know whether IR3mo will have any changes on *ccm1* and *ccm3* mutants on zebrafish heart phenotype.

Within the revised version of the manuscript, we have now included an additional figure which shows the rescue effect of IR3mo treatment on the *krit1^{ly2019c}* mutant cardiac phenotype (Fig EV5A-D). This experiment supports our findings in the murine preclinical model and show that IR3mo has beneficial effects in different vertebrate *ccm* models.

Currently, mutants for the two redundant *ccm3a/pdcd10a* and *ccm3b/pdcd10b* genes of zebrafish have not yet been described and there have been some controversial reports of *ccm3a/b* knockdown phenotypes. Hence, we did not attempt to rescue *ccm3* knockdown phenotypes in zebrafish using IR3mo.

Comment 3. Figure 4b. Similarly it will be relevant to also see the phenotype of IR3mo on CCM2 or PDCD10 siRNA silenced HUVECs.

Within the revised version of the manuscript, we show the effective rescue by IR3mo of *KRIT1*- or *CCM3*-silenced HUVECs, whose phenotype is identical to CCM2-deficient HUVECs (Fig EV5E-J).

Comment 4. Figure 4f. The authors demonstrated the effect of treatment of IR3mo on iCCM2 and iCCM3 mice which alleviated the lesion burden. One might also try to look if there is any effect on lesion burden of IR3mo on iKRIT1 mice model.

We are grateful to the editor for acknowledging that testing IR3mo in a third mouse model would be beyond the scope of this work. Taken together, IR3mo has beneficial effects in different vertebrate *ccm* models.

Referee #3:

The use of multiple animal models for the initial screen, and then a cell culture model for validation of targets makes this a unique study and a very robust study design. The final mouse model, used for only one compound/drug, is important as well as the mouse model is the most relevant for future clinical trials. All in all a very nice and robust study with high scientific and potential medical impact, per my comments below.

Referee #3 (Remarks for Author):

This is a real tour de force in terms of effort and organization, and the study design is quite robust. The use of two different animal models for the drug screen makes this study unique, and the confirmatory cell culture model before moving to the more difficult but clinically relevant mouse model is another nice addition.

I am not an expert regarding the data mining and bioinformatic analyses. Since CCM pathobiology is already extensively studied, it was reassuring that quite a few drugs, drug targets, or pathways/biological processes were identified through the analysis that have already been suggested by other published studies. However, there seem to be so many different and apparently distinct pathways and biological processes that have resulted from the analyses that I would wonder how many are truly relevant to CCM pathobiology? When the analysis points in so many different directions, I have less confidence the relevance of each. However, one advantage to including all the bioinformatic analyses in this paper is that now there are new hypotheses to test concerning CCM pathobiology. So if I consider all this data analyses as hypothesis generating, I am more assured of its importance.

The reviewer points out correctly that a larger number of pathways are misregulated upon loss-of CCM. Hence, some pathways may be changed secondarily after those primarily affected. In future investigations into the molecular control of CCM formation, we will benefit from these pathway lists that provide candidates. Also, in patients, cavernomas are detected only after first clinical manifestations and by that time secondary pathways may be active. Therefore, it is equally relevant to know which are the initial forces driving CCM formation, as well as which are the secondary pathways that are misregulated at later stages of CCM development.

A final point is that IR3mo appears to be a very strong candidate for additional mechanistic studies, but also additional studies in animals that might lead to clinical trials. So the proof of the value of this study is that a new, exciting compound has been identified that might ultimately provide some relief to CCM patients. Congratulations to the investigative team.

We agree with the reviewer that the discovery of IR3mo as an active compound will be beneficial in two ways: first, to further investigate the pathobiology of the CCM disease, and second, as a promising compound (potentially in combination with other compounds) with low toxicity for patients. Currently, we are in the process of establishing a chronic late-onset mouse model of CCM, which will allow us to test the regression of lesions rather than the prevention of their appearance. It will be very interesting to assess whether IR3mo can also be beneficial in this paradigm, which more closely mimicks the human disease.

4th Editorial Decision

7th August 2018

Thank you for the submission of your revised manuscript to EMBO Molecular Medicine, and my apologies for the unusually long review process. We have now received the enclosed reports from the referees. As you will see the reviewers are now supportive, and I am pleased to inform you that we will be able to formally accept your manuscript after the following final editorial amendments:

REFeree REPORTS.

Referee #1 (Remarks for Author):

The manuscript is certainly stronger now, and I think it appropriate for publication.

A small typo: line 433 - models is still -> models are still

Referee #3 (Remarks for Author):

I had no real concerns on the first version, nor on this revision.

Corresponding Author Name: Salim Seyfried
 Journal Submitted to: EMBO Molecular Medicine
 Manuscript Number: EMM-2018-09155-V3